# Blessing of Depth in Linear Regression: Deeper Models Have Flatter Landscape Around the True Solution

**Jianhao Ma**
Department of Industrial & Operations Engineering
University of Michigan
Ann Arbor, MI 48109
`jianhao@umich.edu`

**Salar Fattahi**
Department of Industrial & Operations Engineering
University of Michigan
Ann Arbor, MI 48109
`fattahi@umich.edu`

## Abstract

This work characterizes the effect of depth on the optimization landscape of linear regression, showing that, despite their nonconvexity, deeper models have more desirable optimization landscape. We consider a robust and over-parameterized setting, where a subset of measurements are grossly corrupted with noise, and the true linear model is captured via an $N$-layer diagonal linear neural network. On the negative side, we show that this problem *does not* have a benign landscape: given any $N \geq 1$, with constant probability, there exists a solution corresponding to the ground truth that is neither local nor global minimum. However, on the positive side, we prove that, for any $N$-layer model with $N \geq 2$, a simple sub-gradient method becomes oblivious to such "problematic" solutions; instead, it converges to a balanced solution that is not only close to the ground truth but also enjoys a flat local landscape, thereby eschewing the need for "early stopping". Lastly, we empirically verify that the desirable optimization landscape of deeper models extends to other robust learning tasks, including deep matrix recovery and deep ReLU networks with $\ell_1$-loss.

## 1 Introduction

Supported by the empirical success of deep models in contemporary learning tasks, it is by now a conventional wisdom that "deeper models generalize better" [21, 31, 7]. Indeed, the flurry of recent attempts towards demystifying this phenomenon is a testament to the amount of research it has spawned: from simple linear regression to more complex and nonlinear models, it is shown that deeper models benefit from a range of desirable statistical properties, such as *depth separation* [33, 15, 34, 35], *implicit bias* [19, 2, 10], and *hierarchical learning* [1], to name a few.

Despite the great promise of deeper models—both theoretically and empirically—the effect of depth on their optimization landscape has remained elusive to this day. A recent body of work attempts to characterize the effect of depth on the loss function through the notion of *benign landscape*. Roughly speaking, an optimization problem has a benign landscape if it is devoid of spurious local minima, and its true solutions—i.e., solutions corresponding to the ground truth—coincide with global minima.

36th Conference on Neural Information Processing Systems (NeurIPS 2022).

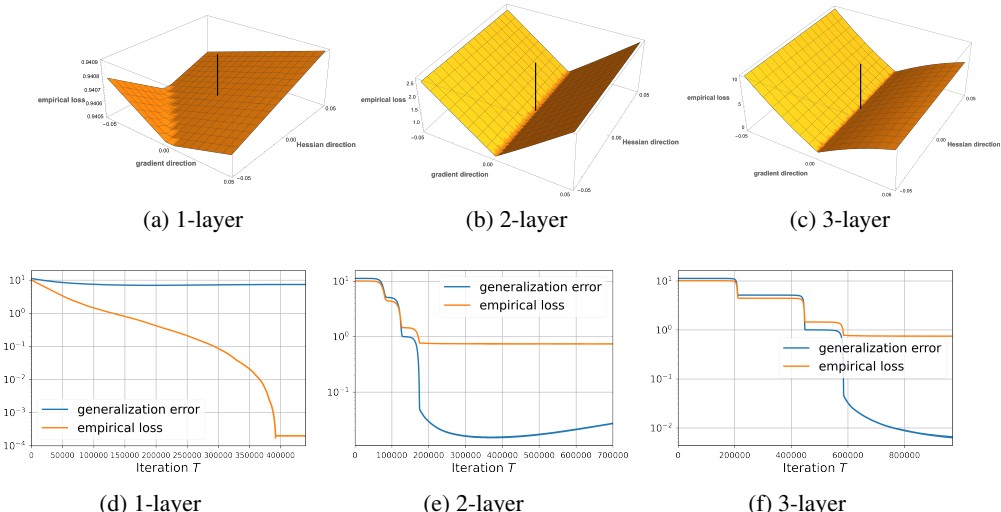

Figure 1: **First row.** Local landscape around the balanced true solution $\mathbf{w}^\star = (\sqrt[N]{\theta^\star}, \ldots, \sqrt[N]{\theta^\star})$ for 1-, 2-, and 3-layer models. $x$ and $y$ axis correspond to the points of the form $\mathbf{w}^\star + \alpha\mathbf{d} + \beta\mathbf{h}$, for different values of $\alpha$ and $\beta$, where $\mathbf{d}$ and $\mathbf{h}$ are respectively the most descent sub-gradient direction and the most negatively curved direction of the Hessian after smoothing. **Second row.** Generalization error and the empirical loss of the solutions found by SubGM for 1-, 2-, and 3-layer models.

It has been shown that 2-layer [5] and multi-layer [23] linear neural networks with nearly-noiseless data have benign landscape. However, the notion of benign landscape is significantly stronger than what is needed in practice. For instance, the existence of spurious local minima may not pose any issue if an optimization algorithm can avoid them efficiently. Another line of research focuses on characterizing the solution trajectory of different local-search algorithms, showing that they enjoy an *implicit bias* that steers them away from undesirable solutions [25, 38, 10, 19, 2, 18]. However, such guarantees only apply to specific trajectories of an algorithm, thereby falling short of any meaningful characterization of the optimization landscape around those trajectories.

## 1.1 Our Contributions

To shed light on the effect of depth on the optimization landscape of deep models, we consider a prototypical problem in machine learning, namely *robust linear regression*, where the goal is to recover a linear model from a limited number of grossly corrupted measurements. Given samples of the form $y_i = \langle \theta^\star, x_i \rangle + \varepsilon_i$, we study the optimization landscape of $\ell_1$-loss under an $N$-layer model defined as $y = f_\mathbf{w}(x) := \langle w_1 \odot w_2 \odot \cdots \odot w_N, x \rangle$. Our results are summarized as follows:

- We prove that, for any $N \geq 1$, there exists at least one true solution that is neither local nor global minimum of $\ell_1$-loss, provided that at least a fraction $p > 0$ of the measurements are corrupted with noise.

- Despite the ubiquity of such "hidden" true solutions, we show that, for any $N$-layer model with $N \geq 2$, a simple sub-gradient method (SubGM) with small initialization converges to a small neighborhood of a balanced true solution. The radius of this neighborhood shrinks with the depth of the model, resulting in more accurate solutions. Moreover, the balancedness of the solution implies that each layer of the model inherits a similar sparsity pattern to the ground truth.

- We prove that deeper models take longer to train, but once trained, the algorithm will stay close to the ground truth for a longer time. This implies that early stopping of the algorithm becomes less crucial for deeper models.

- Finally, we prove that depth flattens the optimization landscape around the solution obtained by SubGM. In particular, we show that, within an $\gamma$-neighborhood of the true solution, the steepest descent direction can reduce the loss by at most $\mathcal{O}(\gamma^N)$, which decreases *exponentially* with $N$.

**Motivating Example.** To showcase our results, we consider an instance of robust linear regression in the over-parameterized setting, where the dimension of $\theta^\star$ is 500 and the number of available samples is 300. Moreover, we assume that 10% of the measurements are corrupted with large noise. The first row of Figure 1 shows the landscape around the balanced ground truth $w_1^\star = \sqrt[N]{\theta^\star}, w_2^\star = \sqrt[N]{\theta^\star}, \ldots, w_N^\star = \sqrt[N]{\theta^\star}$.[1] In particular, $x$ and $y$ axis show the most descent sub-gradient direction and the most negatively curved direction of the Hessian after smoothing.[2] Evidently, there is a sharp transition in the landscape of $N$-layer models: for the 1-layer model, the true solution has strictly negative directions along both sub-gradient and negative curvature. However, these descent directions almost disappear in 2- and 3-layer models. The second row of Figure 1 shows the performance of SubGM on these models. It can be observed that a 1-layer model easily overfits to noise, leading to a vacuous generalization error. On the contrary, a 3-layer model can find a solution that generalizes progressively better than 1- and 2-layer models, demonstrating the algorithmic benefit of the depth.

**Notations:** For two vectors $x, y \in \mathbb{R}^d$, their inner product is defined as $\langle x, y \rangle = x^\top y$, and their Hadamard product is defined as $x \odot y = [x_1 y_1 \cdots x_d y_d]^\top$. For simplicity of notation, we use $\prod_j w_j$ to denote the Hadamard product of $w_1, w_2, \ldots, w_N \in \mathbb{R}^d$. For a vector $x$, $\|x\|$, $\|x\|_\infty$, and $\|x\|_0$ refer to 2-norm, $\infty$-norm, and the number of nonzero elements, respectively. The symbols $a_t \lesssim b_t$ and $a_t = \mathcal{O}(b_t)$ are used to denote $a_t \leq C b_t$, for a universal constant $C$. The notation $a_t = \Theta(b_t)$ is used to denote $a_t = O(b_t)$ and $b_t = \Omega(a_t)$. The $\text{Sign}(\cdot)$ function is defined as $\text{Sign}(x) = x/|x|$ if $x \neq 0$, and $\text{Sign}(0) = [-1, 1]$. We denote $[n] := \{1, 2, \cdots, n\}$. For a vector $x \in \mathbb{R}^d$, we define $x^a = [x_1^a \; x_2^a \; \ldots \; x_d^a]^\top$, for any $a > 0$. In all of our probabilistic arguments, the randomness is only over the input data and noise.

## 2  Problem Formulation

We study the problem of robust and sparse linear regression, where the goal is to estimate a $k$-sparse vector $\theta^\star \in \mathbb{R}^d$ ($k \ll d$) from a limited number of data points $\{(x_i, y_i)\}_{i=1}^m$, where $y_i = \langle \theta^\star, x_i \rangle + \varepsilon_i$, $x_i$ is i.i.d. standard Gaussian vector, and $\varepsilon_i$ is noise. Moreover, for simplicity of our subsequent analysis, we assume that $\theta^\star$ is a non-negative vector. We believe that this assumption can be relaxed without a significant change in our results.

**Assumption 1** (Noise Model). *Given a corruption probability $p$, the noise vector $\varepsilon = [\varepsilon_1 \cdots \varepsilon_m]^\top \in \mathbb{R}^m$ is generated as follows: first, a subset $\mathcal{S} \subset [m]$ with cardinality $pm$ is chosen uniformly at random[3]. Then, for each entry $i \in \mathcal{S}$, the value of $\varepsilon_i$ is drawn independently from a distribution $P_o$, and all the remaining entries are set to zero. Moreover, a random variable $\zeta$ under the distribution $P_o$ satisfies $\mathbb{E}_{P_o}[\zeta] = 0$ and $\mathbb{P}(|\zeta| \geq t_0) \geq p_0$, for some strictly positive constants $t_0$ and $p_0$.*

Our considered noise model does not impose any assumption on the magnitude of the noise or the specific form of its distribution, which makes it particularly suitable for modeling outliers. Note that the assumption $\mathbb{P}(|\varepsilon| \geq t_0) \geq p_0$ is very mild and satisfied for almost all common distributions. Roughly speaking, it implies that the noise takes a nonzero value with a nonzero probability.

To capture the input-output relationship, we consider a class of $N$-layer diagonal linear neural networks of the form $f_\mathbf{w}(x) = \langle w_1 \odot \cdots \odot w_N, x \rangle$, where $\mathbf{w} := (w_1, \cdots, w_N)$ collects the weights of the layers $w_1, \cdots, w_N \in \mathbb{R}^d$. Due to the sparse-and-large nature of the noise, it is natural to minimize the so-called empirical risk with $\ell_1$-loss:

$$\min_\mathbf{w} \mathcal{L}(\mathbf{w}) := \frac{1}{m} \sum_{i=1}^m |f_\mathbf{w}(x_i) - y_i| = \frac{1}{m} \sum_{i=1}^m |\langle w_1 \odot \cdots \odot w_N, x_i \rangle - y_i|. \tag{1}$$

Other variants of empirical risk minimization for linear regression have been studied in the literature. For instance, [10] study the solution trajectory of gradient flow on $\ell_2$-loss, showing that it converges to a solution with the smallest $\ell_1$-norm. Similar analysis has also appeared in more general deep linear neural networks [13, 14]. However, it is well-known that $\ell_2$-loss is highly sensitive to outliers, and $\ell_1$-loss is a better alternative to identify and reject large-and-sparse noise.

---

[1]Later, we will show that a simple sub-gradient method converges to this balanced solution.
[2]To smoothen $|x|$, we replace it with $\sqrt{x^2 + \epsilon}$, for $\epsilon = 10^{-7}$.
[3]Here, for simplicity we assume $pm$ is an integer.

A solution $\bar{\mathbf{w}}$ is called *global* if it corresponds to a global minimizer of $\mathcal{L}(\mathbf{w})$. Moreover, a *local solution* $\bar{\mathbf{w}}$ corresponds to the minimum of $\mathcal{L}(\mathbf{w})$ within an open ball centered at $\bar{\mathbf{w}}$. Finally, a *true solution* $\bar{\mathbf{w}}$ satisfies $\bar{w}_1 \odot \cdots \odot \bar{w}_N = \theta^\star$.

## 3  Main Results

### 3.1  Absence of Benign Landscape

We show that for any arbitrary corruption probability $0 < p < 1/2$ and any number of layers $N \geq 1$, there exists at least one true solution with a strictly negative descent direction, provided that the problem is *over-parameterized*, i.e., $m \lesssim d$.

**Theorem 1** (unidentifiable true solutions). *Define $\mathcal{W} = \{\mathbf{w} : w_1 \odot \cdots \odot w_N = \theta^\star\}$ as the set of all true solutions of an $N$-layer model. For any $N \geq 1$ and $0 < p < 1/2$, the following statements hold:*

*- (Over-parameterized regime) If $m \leq 0.1d$, with probability of at least $1/16$, we have*

$$\inf_{\mathbf{w}^\star \in \mathcal{W}} \inf_{\mathbf{w}:\|\mathbf{w}-\mathbf{w}^\star\|_\infty \leq \gamma} \{\mathcal{L}(\mathbf{w}) - \mathcal{L}(\mathbf{w}^\star)\} \lesssim -p_0 p\gamma, \tag{2}$$

*for any $\gamma \lesssim t_0$.*

*- (Under-parameterized regime) If $m \gtrsim \frac{d}{(1-2p)^2}$, with probability of at least $1 - e^{-\Omega(d)}$, we have*

$$\inf_{\mathbf{w}^\star \in \mathcal{W}} \inf_{\mathbf{w}} \{\mathcal{L}(\mathbf{w}) - \mathcal{L}(\mathbf{w}^\star)\} \geq 0. \tag{3}$$

The above proposition unravels a sharp transition in the landscape of robust linear regression with an $N$-layer model: when $m \lesssim d$, some of the true solutions are likely to be non-critical points, and hence, cannot be recovered via any first-order algorithm. As soon as $m \gtrsim d$, all true solutions become global. This is in stark contrast with the recent results on the benign landscape of robust low-rank matrix recovery with $\ell_1$-loss, which show that, under the so-called *restricted isometry property* (RIP), all the true solutions are global and vice versa [26, 12, 16]. The vector version of RIP is known to hold with $m = \tilde{\Omega}(k)$ samples (see e.g. [3] for a simple proof). Theorem 1 shows that, unlike the low-rank matrix recovery, RIP is *not enough* to guarantee the equivalence between the true and global solutions in deep linear models.

The detailed proof of this theorem can be found in Appendix C.1. Here, we provide a proof sketch for $N = 1$ and $N = 2$ to elucidate the key ideas.

*Proof sketch of Theorem 1.* For 1-layer model, the set of true solutions $\mathcal{W}$ reduces to a singleton $\{\theta^\star\}$. Upon choosing $\mathbf{w} = \theta^\star$, we prove the existence of a perturbation $\|\Delta\theta\| \leq \gamma$ that satisfies $\mathcal{L}(\theta^\star) - \mathcal{L}(\theta^\star + \Delta\theta) < 0$. The perturbed loss takes the following form

$$\mathcal{L}(\theta^\star + \Delta\theta) = \frac{1}{m} \sum_{i \in \bar{\mathcal{S}}} |\langle \Delta\theta, x_i \rangle| + \frac{1}{m} \sum_{i \in \mathcal{S}} |\langle \Delta\theta, x_i \rangle - \varepsilon_i|,$$

Consider the following feasibility problem:

$$\text{find} \quad \alpha \quad \text{s.t.} \quad \langle \alpha, x_i \rangle = 0, \forall i \in \bar{\mathcal{S}}, \quad \langle \alpha, x_i \rangle = \varepsilon_i, \forall i \in \mathcal{S}.$$

Since $m \leq d$ and $\{x_i\}_{i=1}^m$ are i.i.d. standard Gaussian vectors, they are linearly independent almost surely. Moreover, with high probability, at least one of $\varepsilon_i$'s will be nonzero. Therefore, with high probability, the above system of linear equations has at least one nonzero feasible solution. Suppose that $\bar{\alpha} \neq 0$ is one such solution. Define $\Delta\theta = \gamma\bar{\alpha}/\|\bar{\alpha}\|$ for some $0 < \gamma < \|\bar{\alpha}\|$. One can write

$$\mathcal{L}(\theta^\star + \Delta\theta) = \frac{1}{m}\left(1 - \frac{\gamma}{\|\bar{\alpha}\|}\right) \sum_{i \in \mathcal{S}} |\varepsilon_i| < \mathcal{L}(\theta^\star),$$

implying that $\Delta\theta$ is indeed a descent direction. Now, consider a 2-layer model. It is easy to verify the existence of a true solution $\mathbf{w} = (w_1, w_2)$ such that $w_1 \odot w_2 = \theta^\star$ and $\|w_1\|_0 = d$. Consider a perturbation of the form $\Delta\mathbf{w} = (0, \Delta w_2)$. One can write

$$\mathcal{L}(\mathbf{w} + \Delta\mathbf{w}) = \frac{1}{m} \sum_{i \in \bar{\mathcal{S}}} |\langle w_1 \odot \Delta w_2, x_i \rangle| + \frac{1}{m} \sum_{i \in \mathcal{S}} |\langle w_1 \odot \Delta w_2, x_i \rangle - \varepsilon_i|.$$

Since $w_1$ is devoid of zero elements, there exists a nonzero $\Delta w_2$ such that $w_1 \odot \Delta w_2 = \gamma \bar{\alpha}/\|\bar{\alpha}\|$ for $\gamma < \|\bar{\alpha}\|$. Arguments analogous to 1-layer model can then be invoked to show that the constructed perturbation is indeed a descent direction. A similar idea can be naturally extended to $N \geq 3$. $\qquad \square$

Theorem 1 implies that, despite their convexity, 1-layer models are *not* suitable for the robust linear regression since the set of true solutions (which is a singleton $\mathcal{W} = \{\theta^\star\}$) is unidentifiable. However, despite the existence of unidentifiable true solutions in $N$-layer models with $N \geq 2$, we will show that a simple SubGM converges to a *balanced* true solution, even if an arbitrarily large fraction of the measurements are corrupted with arbitrarily large noise values. This further sheds light on the desirable landscape of deeper models in the context of linear regression.

---

**Algorithm 1** Sub-gradient Method

---

**Input:** Data points $\{(x_i, y_i)\}_{i=1}^m$, number of iterations $T$, the initial point $\mathbf{w}_0$, and the step-size $\{\eta^{(t)}\}_{t=0}^T$;
**Output:** Solution $\mathbf{w}^{(T)}$ to (1);
**for** $t \leq T$ **do**
$\quad$ Select a direction $\mathbf{d}^{(t)}$ from the sub-differential $\partial\mathcal{L}(\mathbf{w}^{(t)})$ defined as:

$$\partial_{w_i}\mathcal{L}(\mathbf{w}) = \frac{1}{m}\sum_{j=1}^m \mathrm{Sign}\left(y_j - \left\langle \prod_k w_k, x_j \right\rangle\right) x_j \odot \prod_{k \neq i} w_k; \qquad (4)$$

$\quad$ Set $\mathbf{w}^{(t+1)} \leftarrow \mathbf{w}^{(t)} - \eta^{(t)}\mathbf{d}^{(t)}$;
**end for**

---

### 3.2 Convergence of Sub-gradient Method

At every iteration $t$, SubGM selects a direction $\mathbf{d}^{(t)}$ from the sub-differential of the $\ell_1$-loss (defined as (4)), and updates the solution as $\mathbf{w}^{(t+1)} = \mathbf{w}^{(t)} - \eta^{(t)}\mathbf{d}^{(t)}$; see Algorithm 1 for details. Our next two theorems characterize the performance of SubGM with small initialization on $N$-layer models. We consider the cases $N = 2$ and $N \geq 3$ separately, as SubGM behaves differently on these models. We define $\kappa = \theta^\star_{\max}/\theta^\star_{\min}$ as the condition number, where $\theta^\star_{\max}$ and $\theta^\star_{\min}$ are the maximum and minimum nonzero elements of $\theta^\star$, respectively.

**Theorem 2** (2-layer model). *Consider the iterations of SubGM $\{\mathbf{w}^{(t)}\}_{t=0}^T$ applied to $\mathcal{L}(\mathbf{w})$ with $N = 2$ and step-size $\eta \lesssim 1$. Suppose that the initial point satisfies $w_j^{(0)} = \Theta(\sqrt{\alpha}\mathbf{1})$, $j = 1, 2$, where $0 < \alpha \lesssim d^2 m/k$. Moreover, suppose that $m \gtrsim \frac{k^2\kappa^2\log^2(m)\log(d)\log(\|\theta^\star\|/\alpha)}{(1-p)^2}$. Then, the following statements hold with probability of $1 - Ce^{-\tilde{\Omega}(k)}$:*

- ***Convergence guarantee:*** *After $\frac{1}{\eta}\log\left(\frac{1}{\alpha}\right) \lesssim \bar{T} \lesssim \frac{k^{3/2}}{\eta}\log\left(\frac{1}{\alpha}\right)$ iterations, we have*

$$\left\|w_1^{(\bar{T})} \odot w_2^{(\bar{T})} - \theta^\star\right\| \lesssim \eta\theta^\star_{\max} \vee \sqrt{d^2 m}\alpha^{1-\tilde{\Theta}\left(\frac{k^2}{\sqrt{(1-p)^2 m}}\right)}.$$

- ***Balanced property:*** *For every $0 \leq t \leq \bar{T}$, we have*

$$\left\|w_1^{(t)} - w_2^{(t)}\right\|_\infty \lesssim \alpha^{0.5-\tilde{\Theta}\left(\frac{k^2}{\sqrt{(1-p)^2 m}}\right)}.$$

- ***Long escape time:*** *For every $\bar{T} \leq t \leq \sqrt{\frac{m(1-p)^2}{k}}\bar{T}$, we have*

$$\left\|w_1^{(t)} \odot w_2^{(t)} - \theta^\star\right\| \lesssim \eta\theta^\star_{\max} \vee \sqrt{d^2 m}\alpha^{0.5-\tilde{\Theta}\left(\frac{k^2}{\sqrt{(1-p)^2 m}}\right)}.$$

*Furthermore, if $m \gtrsim d\log(m)/(1-p)^2$, with probability of $1 - Ce^{-\tilde{\Omega}(k)}$ and for every $t \geq \bar{T}$, we have*

$$\left\|w_1^{(t)} \odot w_2^{(t)} - \theta^\star\right\| \lesssim \eta\theta^\star_{\max} \vee \sqrt{d^2 m}\alpha^{1-\tilde{\Theta}\left(\frac{k^2}{\sqrt{m(1-p)^2}}\right)}\left(1 - \Omega\left(\eta/\sqrt{d}\right)\right)^{t-\bar{T}}.$$

We provide the main idea behind the proof of Theorem 2 in Section 4. The formal proof can be found in the appendix. A few observations are in order based on Theorem 2. First, for any $\varepsilon > 0$, SubGM is guaranteed to satisfy $\left\| w_1^{(t)} \odot w_2^{(t)} - \theta^\star \right\| \lesssim \varepsilon$ after $\mathcal{O}((1/\varepsilon)\log(d/\varepsilon))$ iterations, provided that $\eta = \Theta(\varepsilon)$ and $\alpha = \varepsilon^2/(d^2 m)$. Based on our numerical results (provided in the appendix), we believe that it is possible to establish a linear convergence for SubGM with a geometric step-size; a rigorous verification of this conjecture is considered future work. Second, although SubGM converges to a vicinity of a true solution quickly, it will stay there for a significantly longer time—in particular, $\sqrt{m(1-p)^2/k}$ times longer than its initial convergence time. Such behavior is also exemplified in our simulations (see Figure 1e). After this escape time, the algorithm may slowly converge to an *overfitted* solution with a better training loss. Moreover, if $m \gtrsim d$, SubGM will continuously converge to a true solution at an exponential rate, and it will never diverge. Finally, Theorem 2 shows that SubGM implicitly favors balanced solutions, i.e. solutions whose factors have similar magnitudes. Combined with the convergence result of SubGM, we immediately conclude that SubGM converges to a particular solution of the form $(\sqrt{\theta^\star}, \sqrt{\theta^\star})$. Therefore, the solution found by SubGM will enjoy the same (approximate) sparsity pattern as $\theta^\star$.

**Theorem 3** (*N-layer models*). *Consider the iterations of SubGM $\{\mathbf{w}^{(t)}\}_{t=0}^T$ applied to $\mathcal{L}(\mathbf{w})$ with $N \geq 3$ and step-size $\eta \lesssim N^{-1}\kappa^{-\frac{N-2}{N}}$. Suppose that the initial point satisfies $w_j^{(0)} = \Theta(\alpha^{1/N}\mathbf{1})$, where $0 < \alpha \lesssim d^2 m/k$. Moreover, suppose that $m \gtrsim \frac{k^2\kappa^4 \log^2(m)\log(d)\log(\|\theta^\star\|/\alpha)}{(1-p)^2}$. Then, the following statements hold with probability of $1 - Ce^{-\tilde{\Omega}(k)}$:*

- ***Convergence guarantee:*** *After $\frac{1}{N\eta}\alpha^{-\frac{N-2}{N}} \lesssim \bar{T} \lesssim \frac{k^{3/2}}{N\eta}\alpha^{-\frac{N-2}{N}}$ iterations, we have*

$$\left\| \prod w_i^{(\bar{T})} - \theta^\star \right\| \lesssim N\eta\theta_{\max}^\star \vee \sqrt{d^2 m}\alpha.$$

- ***Balanced property:*** *For every $0 \leq t \leq \bar{T}$, we have*

$$\left| w_{i,l}^{(t)} - w_{j,l}^{(t)} \right| = \mathcal{O}\left(\alpha^{1/N}\right), \qquad \text{for } 1 \leq i < j \leq N, l : \theta_l^\star = 0,$$
$$\left| w_{i,l}^{(t)} - w_{j,l}^{(t)} \right| = \tilde{\mathcal{O}}\left(\sqrt[N]{\theta_l^\star}\sqrt{k^3/m}\right), \quad \text{for } 1 \leq i < j \leq N, l : \theta_l^\star \neq 0.$$

- ***Long escape time:*** *For every $\bar{T} \leq t \leq \sqrt{\frac{m(1-p)^2}{k}}\bar{T}$, we have*

$$\left\| \prod w_i^{(t)} - \theta^\star \right\| \lesssim N\eta\theta_{\max}^\star \vee \sqrt{d^2 m\alpha},$$

*Furthermore, if $m \gtrsim d^{\frac{2N-2}{N}}\log(m)/(1-p)^2$, with probability of at least $1 - Ce^{-\tilde{\Omega}(k)}$ and for every $t > \bar{T}$, we have*

$$\left\| \prod w_i^{(t)} - \theta^\star \right\| \lesssim N\eta\theta_{\max}^\star \vee \left( \frac{\sqrt{d^2 m}\alpha}{\sqrt{d^2 m}\alpha N\eta d^{-\frac{N-1}{N}}(t-\bar{T})+1} \right)^{\frac{N}{N-2}}.$$

The proof of this theorem can be found in the appendix. Theorem 3 sheds light on an important benefit of $N$-layer models with $N \geq 3$ compared to 2-layer models: for sufficiently small step-size, deeper models improve the generalization error by a factor of $(1/\alpha)^{\tilde{\Theta}\left(k^2/\sqrt{((1-p)^2 m)}\right)}$. This improvement is particularly significant when both $\alpha$ and $m$ are small. However, such improvement comes at the expense of a slower convergence rate. In particular, after setting $\eta = \Theta(\varepsilon/N)$, and $\alpha = \varepsilon/\sqrt{d^2 m}$, SubGM needs $\mathcal{O}\left((1/\varepsilon)^{1+\frac{N-2}{N}}\right)$ iterations to obtain an $\epsilon$-accurate solution. Evidently, the convergence rate deteriorates with $N$, ultimately approaching $\mathcal{O}\left(1/\varepsilon^2\right)$ for infinitely deep models. This can be observed in practice: Figures 1e and 1f show that 3-layer model enjoys a better generalization error compared to 2-layer model, but suffers from a slower convergence rate. This slower convergence rate also manifests itself in a more stable behavior of the algorithm: for deeper models, SubGM stays close to the ground truth for a longer time. Finally, the balanced property of the solution obtained via SubGM extends to $N$-layer models. In particular, SubGM converges to a particular solution of the form $(\sqrt[N]{\theta^\star}, \ldots, \sqrt[N]{\theta^\star})$, thereby inheriting the same sparsity pattern as $\theta^\star$.

### 3.3 Local Landscape Around Balanced Solution

In the previous section, we showed that SubGM converges to a balanced solution. In this section, we characterize the local landscape around this balanced solution, proving that it becomes flatter for deeper models.

**Theorem 4** (flatness around balanced solution). *Suppose that $k \log(d)/(1 - 2p)^2 \lesssim m \leq 0.1d$ and $p < 1/2$. Let $\mathbf{w}^\star = (\sqrt[N]{\theta^\star}, \ldots, \sqrt[N]{\theta^\star})$. Then, for any $N \geq 2$ and $\gamma \leq t_0/\sqrt{d} \wedge 1$, the following statements hold:*

- *With probability at least $1 - e^{-\Omega(k)}$, we have*
$$\inf_{\mathbf{w}:\|\mathbf{w}-\mathbf{w}^\star\|_\infty \leq \gamma} \{\mathcal{L}(\mathbf{w}^\star) - \mathcal{L}(\mathbf{w})\} \gtrsim -\frac{d}{\sqrt{m}}\gamma^N.$$

- *With probability at least $1/16$, we have*
$$\inf_{\mathbf{w}:\|\mathbf{w}-\mathbf{w}^\star\|_\infty \leq \gamma} \{\mathcal{L}(\mathbf{w}^\star) - \mathcal{L}(\mathbf{w})\} \lesssim -\sqrt{p_0 p}\frac{d}{\sqrt{m}}\gamma^N.$$

Theorem 4 shows that, within a $\gamma$-neighborhood of $\mathbf{w}^\star$, the most descent direction from $\mathbf{w}^\star$ can reduce the loss by at most $\mathcal{O}\left(d/\sqrt{m} \cdot \gamma^N\right)$, which decreases exponentially with $N$. Moreover, in the noisy setting, the above theorem implies that $\mathbf{w}^\star$ is likely to be neither local nor global minimum since it has a descent direction. However, the flatness of the landscape around $\mathbf{w}^\star$ enables SubGM to stay close to the balanced solution for a long time.

**Remark 1.** *Note that the choice of $\ell_\infty$-ball for the perturbation set is to ensure that the size of the possible perturbations per layer remains independent of the depth of the model. This is indeed crucial to ensure a fair comparison between models with different depths: alternative choices of the perturbation set, such as $\ell_q$-ball with $1 \leq q < \infty$ (e.g. $\ell_2$-ball) would shrink the size of the feasible per-layer perturbations with $N$, thereby leading to an unfair advantage to deeper models.*

## 4 Proof Techniques

At the crux of our proof technique for Theorems 2 and 3 lies the following decomposition of the sub-differential:
$$\partial\mathcal{L}(\mathbf{w}) = \underbrace{\xi \cdot \partial\bar{\mathcal{L}}(\mathbf{w})}_{\text{expected subdiff.}} + \underbrace{\left(\partial\mathcal{L}(\mathbf{w}) - \xi \cdot \partial\bar{\mathcal{L}}(\mathbf{w})\right)}_{\text{subdiff. deviation}}, \quad \text{for some strictly positive } \xi.$$

In the above decomposition, $\bar{\mathcal{L}}(\mathbf{w})$ is called *expected loss*, and is defined as $\bar{\mathcal{L}}(\mathbf{w}) = \|w_1 \odot \cdots \odot w_N - \theta^\star\|$. As will be shown later, $\bar{\mathcal{L}}(\mathbf{w})$ captures the expected behavior of the empirical loss $\mathcal{L}(\mathbf{w})$. To analyze the behavior of SubGM on $\mathcal{L}(\mathbf{w})$, we first consider the ideal scenario, where $\mathcal{L}(\mathbf{w})$ coincides with its expectation. Then, we extend our analysis to the general case by controlling the sub-differential deviation. In particular, we show that the desirable convergence properties of SubGM extends to $\mathcal{L}(\mathbf{w})$, provided that its sub-differentials are "direction-preserving", i.e., $\mathbf{d} \approx \xi\bar{\mathbf{d}}$, for every $\mathbf{d} \in \partial\mathcal{L}(\mathbf{w}), \bar{\mathbf{d}} \in \partial\bar{\mathcal{L}}(\mathbf{w})$ and some $\xi > 0$. To formalize this idea, we first provide a more concise characterization of $\partial\mathcal{L}(\mathbf{w})$:

$$\partial_{w_i}\mathcal{L}(\mathbf{w}) = \left\{q \odot \prod_{k \neq i} w_k : q \in \mathcal{Q}\left(\theta^\star - \prod_k w_k\right)\right\}, \quad \text{where } \mathcal{Q}(z) = \frac{1}{m}\sum_{i=1}^m \text{Sign}\left(\langle x_i, z\rangle + \varepsilon_i\right)x_i.$$

**Definition 1** (approximately sparse vectors). *We say a vector $v \in \mathbb{R}^d$ is $(k, \vartheta)$-approximately sparse if there exists a vector $u$, such that $\|u\|_0 \leq k$ and $\|u - v\| \leq \vartheta$.*

**Proposition 1** (direction-preserving property). *Suppose that $m \gtrsim \frac{k \log^2(m) \log(d) \log(R/\vartheta)}{(1-p)^2 \delta^2}$ for some $R, \vartheta, \delta > 0$. Then, with probability of at least $1 - Ce^{-\Omega(m\delta^2)}$, the following inequality holds for any $q \in \mathcal{Q}(z)$ and any $(k, \vartheta)$-approximately sparse vector $z$ that satisfies $\sqrt{dm/k}\vartheta \log(1/\vartheta) \lesssim \|z\| \leq R$:*

$$\left\|q - \sqrt{\frac{2}{\pi}}\left(1 - p + p\mathbb{E}\left[e^{-\varepsilon^2/(2\|z\|)}\right]\right)\frac{z}{\|z\|}\right\|_\infty \leq \delta. \tag{5}$$

*Moreover, if $m \gtrsim \frac{d \log(m)}{(1-p)^2 \delta^2}$, with probability of $1 - Ce^{-\Omega(m\delta^2)}$, (5) holds for every $z \in \mathbb{R}^d$.*

Proposition 1 is analogous to *Sign-RIP* condition introduced in [29, 28] for the robust low-rank matrix recovery, and is at the heart of our proofs for Theorems 2 and 3. Suppose that $\theta^\star - \prod_k w_k$ is a $(k, \vartheta)$-approximately sparse and satisfies (5). Then, we have $\left\| \mathbf{d} - \bar{\mathbf{d}} \right\|_\infty \le \left( \max_i \left\{ \prod_{k \ne i} w_k \right\} \right) \delta$, which in turn provides an upper bound on the sub-differential deviation.

## 4.1 Proof Sketch of Theorem 2

To streamline the presentation, here we only provide simplified versions of our key ideas, which inevitably lead to looser guarantees. To streamline the proof, we assume that $\theta_1^\star \ge \cdots \ge \theta_k^\star > \theta_{k+1}^\star = \cdots = \theta_d^\star = 0$. Moreover, for simplicity of notation, we denote $u = w_1$ and $v = w_2$. Consider the following decomposition:

$$u \odot v = [\underbrace{u_1 v_1 \ldots u_k v_k}_{S} \ \underbrace{u_{k+1} v_{k+1} \ldots u_d v_d}_{E}]^\top. \tag{6}$$

The vectors $S$ and $E$ are called *signal* and *residual terms*, respectively. Evidently, we have $u \odot v = \theta^\star$ if and only if $S = [\theta_1^\star, \ldots, \theta_k^\star]^\top$ and $E = 0$. Based on this observation, our goal is to show that the signal term converges to $[\theta_1^\star, \ldots, \theta_k^\star]^\top$ exponentially fast, while the error term remains small throughout the solution trajectory.

**Lemma 1** (signal dynamic; informal)**.** *Suppose that* (5) *holds for* $z = \theta^\star - u^{(t)} \odot v^{(t)}$, *and* $\left\| \theta^\star - u^{(t)} \odot v^{(t)} \right\| \gtrsim \eta \left\| \theta^\star \right\|$. *Then, we have*

$$u_i^{(t+1)} v_i^{(t+1)} \ge \left( 1 + 2\eta \left( \frac{\theta_i^\star - u_i^{(t)} v_i^{(t)}}{\left\| u^{(t)} \odot v^{(t)} - \theta^\star \right\|} + \delta_i \right) \right) u_i^{(t)} v_i^{(t)}, \tag{7}$$

*for some* $|\delta_i| \le \delta$ *and every* $i = 1, \ldots, k$.

**Lemma 2** (residual dynamic; informal)**.** *Suppose that* (5) *holds for* $z = \theta^\star - u^{(t)} \odot v^{(t)}$, *and* $\left\| \theta^\star - u^{(t)} \odot v^{(t)} \right\| \gtrsim \eta \left\| \theta^\star \right\|$. *Then, we have*

$$\left( u_i^{(t+1)} \right)^2 + \left( v_i^{(t+1)} \right)^2 \le (1 + \mathcal{O}(\eta\delta)) \left( \left( u_i^{(t)} \right)^2 + \left( v_i^{(t)} \right)^2 \right), \tag{8}$$

*for every* $i = k+1, \ldots, d$.

**Lemma 3** (difference dynamic; informal)**.** *Suppose that* (5) *holds for* $z = \theta^\star - u^{(t)} \odot v^{(t)}$, *and* $\left\| \theta^\star - u^{(t)} \odot v^{(t)} \right\| \gtrsim \eta \left\| \theta^\star \right\|$. *Then, we have*

$$u_i^{(t+1)} - v_i^{(t+1)} = \left( u_i^{(t)} - v_i^{(t)} \right) \left( 1 - \eta \frac{\theta_i^\star - u_i^{(t)} v_i^{(t)}}{\left\| u^{(t)} \odot v^{(t)} - \theta^\star \right\|} + \eta\delta_i \right), \tag{9}$$

*for some* $|\delta_i| \le \delta$ *and every* $i = 1, \ldots, d$.

**Convergence guarantee.** For any fixed $i = 1, \ldots, k$, we show that $u_i^{(t)} v_i^{(t)} = \theta_i^\star \pm \mathcal{O}(\delta) \left\| \theta^\star \right\|$ after $\mathcal{O}(\left\| \theta^\star \right\| / (\eta\theta_i^\star) \log(1/\alpha))$ iterations. To see this, suppose that $T_i$ is the largest iteration such that $u_i^{(t)} v_i^{(t)} \le \theta_i^\star$ for every $t \le T_i$. Moreover, suppose that $\left\| u^{(t)} \odot v^{(t)} \right\| \le C \left\| \theta^\star \right\|$, for sufficiently large $C$ (this is proven in the appendix). Under these assumptions, (7) reduces to

$$u_i^{(t+1)} v_i^{(t+1)} \ge \left( 1 + \Omega(1) \frac{\eta\theta_i^\star}{\left\| \theta^\star \right\|} \right) u_i^{(t)} v_i^{(t)}. \tag{10}$$

which implies that $T_i \lesssim \left\| \theta^\star \right\| / (\eta\theta_i^\star) \log(1/\alpha)$. For any $t > T_i$, define $y_i^{(t)} = \theta_i^\star - u_i^{(t)} v_i^{(t)}$. One can write

$$y_i^{(t+1)} \le \left( 1 - \Omega(1) \frac{\eta\theta_i^\star}{\left\| \theta^\star \right\|} \right) y_i^{(t)} + \eta\delta\theta_i^\star. \tag{11}$$

Hence, with additional $\mathcal{O}\left( \left\| \theta^\star \right\| / (\eta\theta_1^\star) \right)$ iterations, we have $u_i^{(t)} v_i^{(t)} = \theta_i^\star \pm \mathcal{O}(\delta) \left\| \theta^\star \right\|$. On the other hand, Lemma 2 implies that, for any $i = k+1, \ldots, d$ and $t \lesssim \left\| \theta^\star \right\| / (\eta\theta_k^\star) \log(1/\alpha)$, we have

$$\left( u_i^{(t)} \right)^2 + \left( v_i^{(t)} \right)^2 \lesssim \alpha \left( 1 + \mathcal{O}(\eta\delta) \right)^{\mathcal{O}\left( \frac{\left\| \theta^\star \right\|}{\eta\theta_k^\star} \log\left( \frac{1}{\alpha} \right) \right)} \lesssim \alpha^{1 - \mathcal{O}(\sqrt{k}\kappa\delta)},$$

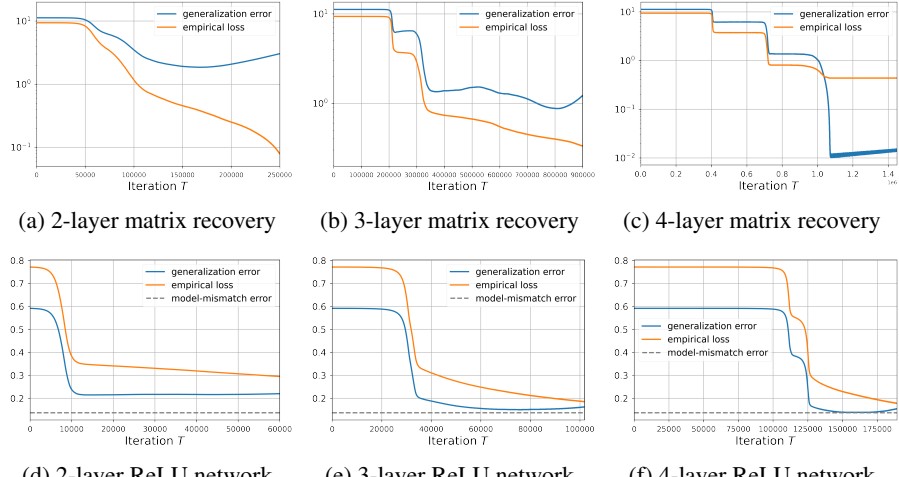

(a) 2-layer matrix recovery  (b) 3-layer matrix recovery  (c) 4-layer matrix recovery

(d) 2-layer ReLU network  (e) 3-layer ReLU network  (f) 4-layer ReLU network

Figure 2: **Deep matrix recovery (first row).** The ground truth $X^\star \in \mathbb{R}^{20 \times 20}$ with $\text{rank}(X^\star) = 3$ is chosen randomly. The elements of the measurement matrices are selected from $\mathcal{N}(0, 1)$, and the sample size is set to $m = 180$. The corruption probability is set to $p = 0.05$ with distribution $\mathcal{N}(0, 100)$. We use SubGM with step-size $\eta = 0.001$ and Gaussian initialization with an initialization scale $\alpha = 1 \times 10^{-3}$. **ReLU models (second row).** The samples are chosen from $y_i = \sin(\theta^{\star \top} x_i) + \varepsilon_i$ where $\theta^\star \in \mathbb{R}^{50}$ is randomly generated with $\|\theta^\star\|_0 = 2$, $x_i \sim \mathcal{N}(0, I_{50})$, and $\varepsilon_i \sim \mathcal{N}(0, 25)$ with corruption probability $p = 0.05$. The sample size is set to be $m = 1500$. We use SubGM with step-size $\eta = 0.001$ and Gaussian initialization $\mathcal{N}(0, \alpha^{2/N}/d)$.

where $\kappa = \theta_1^\star / \theta_k^\star$ is the condition number of $\theta^\star$. Combining the above dynamics, we have

$$\left\| u^{(t)} \odot v^{(t)} - \theta^\star \right\| \lesssim \eta \left\| \theta^\star \right\| \vee \sqrt{k} \left\| \theta^\star \right\| \delta \vee \sqrt{d} \alpha^{1 - \mathcal{O}\left(\sqrt{k} \kappa \delta\right)}.$$

In the appendix, we provide a more refined analysis that relaxes the dependency of the final error on $\delta$ and $\kappa$.

**Long escape time.** We show in the appendix that after the first stage, the residual becomes the dominant term in the final error. This together with Lemma 2 implies that, for every $t \lesssim \frac{\|\theta^\star\|}{\eta \theta_k^\star \sqrt{\delta}} \log(1/\alpha)$, we have $\|E\| \lesssim \sqrt{d} \alpha^{1 - \sqrt{k} \kappa \sqrt{\delta}}$.

**Balanced property.** We have $u_i^{(t)} v_i^{(t)} \leq \theta_i^\star$ for every $i \in [k]$, and $|u_i^{(t)} v_i^{(t)}| \lesssim \alpha^{1 - \mathcal{O}(\sqrt{k} \kappa \delta)}$ for every $i = k + 1, \ldots, d$. Therefore, Lemma 3 can be invoked to verify $\left| u_i^{(t+1)} - v_i^{(t+1)} \right| \leq (1 + \mathcal{O}(\eta \delta)) \left| u_i^{(t)} - v_i^{(t)} \right|$. This in turn leads to

$$\left| u_i^{(t)} - v_i^{(t)} \right| \lesssim \sqrt{\alpha} \left(1 + \mathcal{O}(\eta \delta)\right)^{\mathcal{O}\left(\frac{\|\theta^\star\|}{\eta \theta_k^\star} \log\left(\frac{1}{\alpha}\right)\right)} \lesssim \alpha^{0.5 - \mathcal{O}(\sqrt{k} \kappa \delta)}.$$

## 5 Numerical Experiments: Beyond Linear Regression

In this section, we empirically verify that the benefits of depth extend to the robust variants of deep matrix recovery and ReLU networks with $\ell_1$-loss.

**Deep Matrix Recovery.** In low-rank matrix recovery, the goal is to recover a low-rank matrix $X^\star \in \mathbb{R}^{d \times d}$, from a limited number of noisy measurements of the form $y_i = \langle A_i, X^\star \rangle + \varepsilon_i$. To recover $X^\star$, we consider a deep factorized model of the form $W_1 W_2 \ldots W_N$, where $W_i \in \mathbb{R}^{d \times d}$ for $i = 1, \ldots, N$, and minimize the $\ell_1$-loss $(1/m) \sum_{i=1}^m |y_i - \langle A_i, W_1 W_2 \cdots W_N \rangle|$ via SubGM. When $N = 2$, the above model reduces to the famous Burer-Monteiro approach [8]. We assume that $5\%$ of the measurements are grossly corrupted with noise. The first row of Figure 2 shows the performance of SubGM on 2-, 3-, and 4-layer models. It can be seen that the 4-layer model outperforms shallower models, achieving a generalization error that is proportional to the step-size.

**Deep ReLU Network on Synthetic Dataset.** As another experiment, we analyze the effect of depth on the performance of SubGM with ReLU networks and $\ell_1$-loss. Given an input $x \in \mathbb{R}^d$, the output of an $N$-layer ReLU network is defined as $f_{\mathbf{W}}(x) = W_N \sigma (W_{N-1} \cdots \sigma (W_1 x) \cdots)$, where $W_1 \in \mathbb{R}^{m \times d}, W_2, \cdots, W_{N-1} \in \mathbb{R}^{m \times m}$, and $W_N \in \mathbb{R}^{1 \times m}$. Moreover, $\sigma(x) = \max\{0, x\}$ is the ReLU activation function. Given the true function $f^\star(x) = \sin(\theta^{\star\top} x)$, our goal is to train a ReLU model to approximate $f^\star$ as accurately as possible. To this goal, we minimize the $\ell_1$-loss $(1/m) \sum_{i=1}^m |y_i - f_{\mathbf{W}}(x_i)|$. The second row of Figure 2 illustrates the performance of SubGM. It is worth noting that, unlike robust linear regression and deep matrix recovery, there always exists a non-diminishing model-mismatch error between the true and considered ReLU model (shown as a dashed line). Nonetheless, SubGM can achieve this model-mismatch error on a 4-layer ReLU model with only 1500 samples, even if $5\%$ of the measurements are corrupted with large noise.

**Deep ReLU Network on CIFAR Dataset.** We verify that the desirable performance of SubGM with $\ell_1$-loss can be extended to its stochastic variant with mini-batches on CIFAR-10 and CIFAR-100 [24], outperforming cross-entropy (CE) loss, which is considered as one of the most suitable loss functions for CIFAR datasets. To show this, we use standard ResNet architectures [21] with $\ell_1$-loss and compare it with the cross-entropy loss on noisy CIFAR datasets, where we randomize the labels of $10\%$ of the training dataset. For CIFAR-100 experiment, we use the "loss scaling" trick introduced in [22]. The training details are deferred to Section B.3. The best test accuracy for both CIFAR-10 and CIFAR-100 is reported in Table 1. One can see that $\ell_1$-loss outperforms cross-entropy loss significantly, demonstrating that our framework may be extended to more realistic settings. Moreover, we do observe that the deeper model performs better on CIFAR-100, which aligns with our theoretical result. Based on our simulations, an interesting and important future direction would be to study the optimization landscape of $\ell_1$-loss with more general neural network architectures.

|  | CIFAR-10 | | | CIFAR-100 | | |
| --- | --- | --- | --- | --- | --- | --- |
|  | ResNet-18 | ResNet-34 | ResNet-50 | ResNet-18 | ResNet-34 | ResNet-50 |
| CE loss | 91.52% | 91.53% | 90.87% | 70.17% | 71.22% | 71.30% |
| $\ell_1$-loss | **94.16**% | **93.13**% | **92.68**% | **73.14**% | **74.86**% | **76.46**% |

Table 1: Test accuracy for ResNets on CIFAR-10 and CIFAR-100 datasets with $10\%$ label noise.

## 6 Conclusion

Modern problems in machine learning are naturally nonconvex but can be solved reasonably well in practice. To explain this, a recent body of work has postulated that many optimization problems in machine learning are "convex-like", i.e., they are devoid of spurious local minima. Our work shows that such global property is too restrictive to hold even in the context of linear regression, and instead propose a more refined *trajectory analysis* to better capture the landscape of the problem around the solution trajectory. We show that convex models may be fundamentally ill-suited for linear models, and deeper models–despite their nonconvexity–have provably better optimization landscape around the solution trajectory. Empirically, we show that our analysis may extend beyond linear regression; formal verification of this conjecture is considered an enticing challenge for future research.

## Acknowledgements

We thank Richard Y. Zhang, Cédric Jósz, and Tiffany Wu for helpful discussions and feedback. We thank Ruiqi Gao and Jiaye Teng for insightful discussions in the initial phase of this work. We are also thankful to an anonymous reviewer for pointing out the relationship between the perturbation ball and the depth of linear models. This research is supported, in part, by NSF Award DMS-2152776, ONR Award N00014-22-1-2127, MICDE Catalyst Grant, MIDAS PODS grant, and Startup funding from the University of Michigan.

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
