# Contents

## A   Related Works

**Deep models:**   It is known that deeper models enjoy better *approximation power*. For instance, [15, 34, 35] introduce several functions that are expressible by deep models of moderate size; yet, they cannot be approximated via any shallow network of sub-exponential size. A recent work [33] shows that depth separation may lead to optimization separation. In other words, functions that can be expressed by deeper models can also be efficiently learned via gradient descent. Another line of work shows that deep linear models have a strong implicit bias towards the true solution [19, 2, 10], and that they benefit from *incremental learning* [18, 27]. More generally, [1] show that stochastic gradient descent on deep nonlinear models can provably learn certain complex functions by automatically decomposing them into a series of simpler ones.

**Robust and sparse linear regression:**   Robust and sparse linear regression is a classical problem in statistics, with a wide range of applications in signal and image processing. Regularized methods, including Lasso [36, 9, 6, 32], Best Subset Selection [30, 4], and Forward and Backward Step-wise Regression [17], are considered as most widely used methods for solving robust and sparse linear regression that come equipped with strong statistical and computational guarantees. Recently, sparse linear regression has been used to explore the implicit bias of different optimization algorithms and initialization regimes. [38, 41] show that, for some unregularized overparameterized models, gradient descent (with early stopping) for sparse linear regression achieves minimax error rate. Moreover, [40] study how the scale of the initial point controls the transition between the "kernel" (lazy training) and "rich" regimes, and their corresponding generalization performance. [20] use this problem setting to explore the role of label noise in stochastic gradient descent.

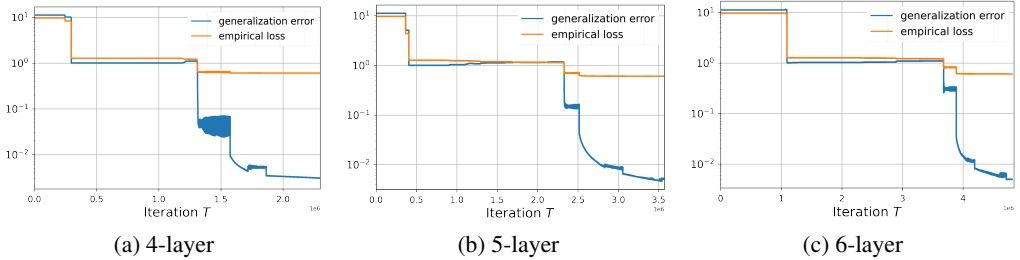

(a) 4-layer            (b) 5-layer            (c) 6-layer

Figure 3: The optimization trajectories of deep models ($N = 4, 5, 6$).

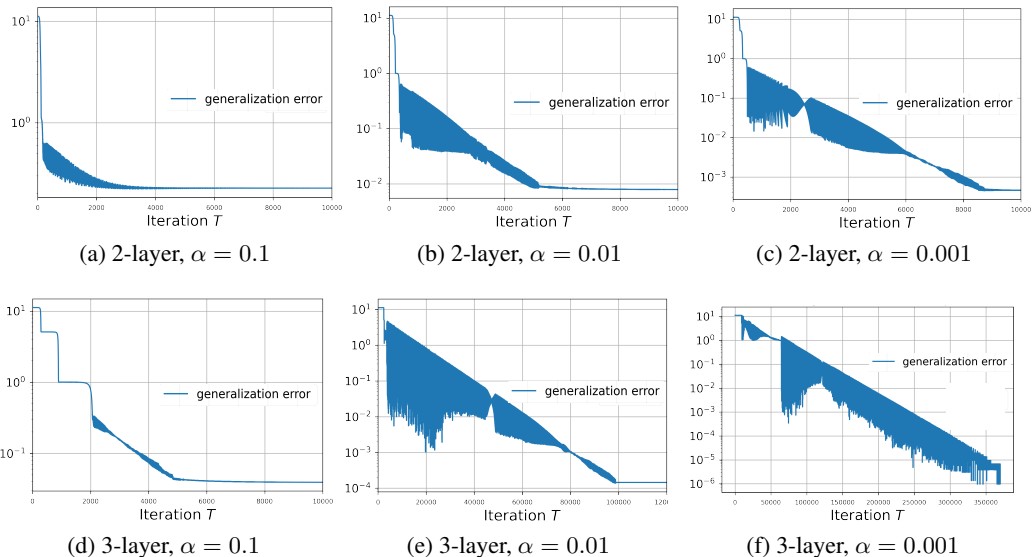

(a) 2-layer, $\alpha = 0.1$      (b) 2-layer, $\alpha = 0.01$      (c) 2-layer, $\alpha = 0.001$

(d) 3-layer, $\alpha = 0.1$      (e) 3-layer, $\alpha = 0.01$      (f) 3-layer, $\alpha = 0.001$

Figure 4: The optimization trajectories of 2 and 3-layer models with different initialization size $\alpha = 0.1, 0.01, 0.001$ using exponentially decayed step-size.

## B  Additional Experiments

In this section, we provide additional experiments on the performance of SubGM on deep models. Our goal is to verify our theoretical results and show the benefits of both small initialization and geometric step-size. Moreover, we show that the desirable performance of SubGM can be observed in its stochastic variant, as well as for different architectures of ResNets with $\ell_1$-loss, and more realistic CIFAR-10 dataset. All simulations are run on a desktop computer with an Intel Core i9 3.50 GHz CPU and 128GB RAM. The reported results are for implementation in Python.

### B.1  Deeper Linear Models

In this experiment, we study the performance of SubGM on deeper models ($N = 4, 5, 6$). To accelerate the training process, we first use a large step-size $\eta_1 = 1 \times 10^{-3}$, and then progressively apply smaller step-sizes $\eta_2 = 1 \times 10^{-4}$ and $\eta_3 = 1 \times 10^{-5}$ as the training loss continues to decay. As shown in Figure 3, deeper models share similar generalization error, outperforming 1- and 2-layer models presented in Figure 1.

### B.2  Geometric Step-size

As shown in our simulations, training $N$-layer models may require millions of iterations even on a small synthetic dataset. As proven in Theorem 3, the training process may become even slower for deeper models. In this experiment, we explore the performance of geometric (i.e., exponentially

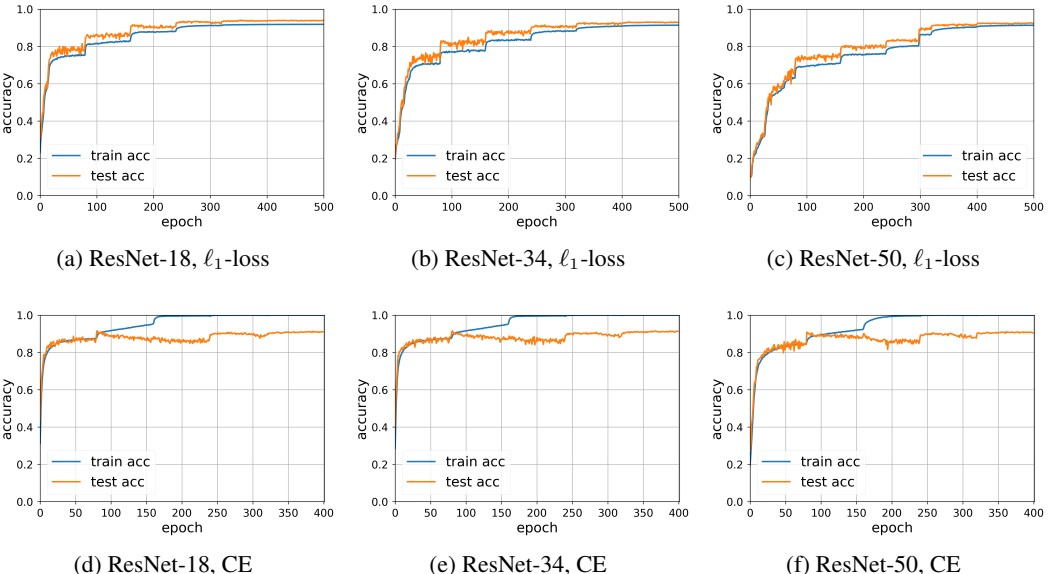

(a) ResNet-18, $\ell_1$-loss      (b) ResNet-34, $\ell_1$-loss      (c) ResNet-50, $\ell_1$-loss

(d) ResNet-18, CE      (e) ResNet-34, CE      (f) ResNet-50, CE

Figure 5: We apply ResNet-18, 34, 50 on noisy CIFAR-10 with both $\ell_1$-loss and cross-entropy loss (CE). For the training dataset, we randomly choose $10\%$ samples and replace their labels with uniform random labels. We use SGD with initial learning rate $\eta = 0.1$, momentum $0.9$, batch size $B = 32$. For every $80$ epochs, we decay the learning rate by a factor $0.33$. We use standard data augmentation. The initialization is set by default in PyTorch.

decaying) step-size on the same dataset. SubGM with a geometric step-size has been widely used for the optimization of $\ell_1$-loss [26, 28], and more general sharp weakly convex functions [11]. Figure 4 shows that a geometric steps-size can lead to a 1000-fold reduction in the required number of iterations. Moreover, a geometric step-size improves the convergence rate to linear. The theoretical justification of this improvement is left as an enticing challenge for future research. Finally, it can be observed that SubGM with geometric step-size performs surprisingly well on deeper models, achieving a generalization error in the order of $10^{-6}$. This further supports the benefits of depth.

## B.3    Experiments on CIFAR Dataset

In this section, we provide the training details for the experiments on both CIFAR-10 and CIFAR-100 where $10\%$ of the training data points are randomly labeled. For CIFAR-100 experiment, we use the "loss scaling" trick introduced in [22]. In particular, we denote the neural network by $f_\theta : \mathbb{R}^d \to \mathbb{R}^C$, where $d$ is the input dimension and $C$ is the number of class. The standard $\ell_1$-loss for the one-hot encoded label vector can be written (at a single point) as

$$\ell = |f_\theta(x)[c] - 1| + \sum_{c' \neq c} |f_\theta(x)[c']| . \tag{12}$$

Here $c$ is the position of the label and $f_\theta(x)[i]$ is the $i$-th coordinate of the prediction. The rescaled $\ell_1$-loss is defined by two parameters $k$ and $M$ as follows:

$$\ell_{\text{scaling}} = k \cdot |f_\theta(x)[c] - M| + \sum_{c' \neq c} |f_\theta(x)[c']| . \tag{13}$$

In our simulation, we choose $k = 5$, and $M = 2$. Moreover, in the CIFAR-100 experiment, we use $\ell_2$-loss with the same $k, M$ in the first 30 epochs as the warm-up phase. The evolution of the training and test accuracy for CIFAR-10 and CIFAR-100 with both $\ell_1$ and CE losses are shown in Figures 5 and 6.

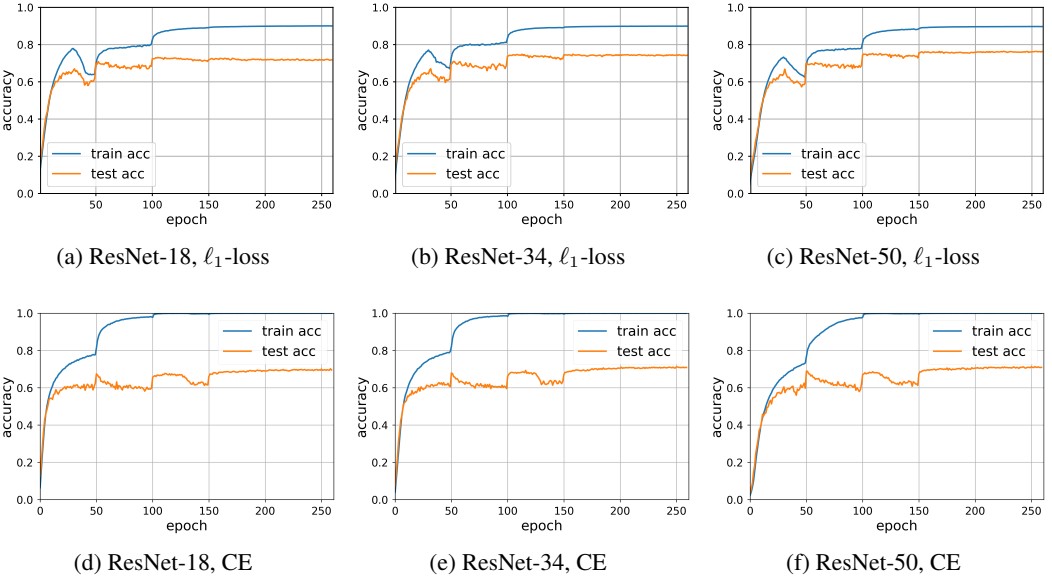

(a) ResNet-18, $\ell_1$-loss      (b) ResNet-34, $\ell_1$-loss      (c) ResNet-50, $\ell_1$-loss

(d) ResNet-18, CE      (e) ResNet-34, CE      (f) ResNet-50, CE

Figure 6: We apply ResNet-18, 34, 50 on noisy CIFAR-100 with both $\ell_1$-loss and cross-entropy loss (CE). The data generator and optimizer are the same as those in CIFAR-10 experiment. For each trial, we set the initial learning rate to be $\eta = 0.05$ and run 260 epochs and decay the learning rate with factor 0.33 for each 50 epochs.

# C Proofs of Landscape Analysis

## C.1 Proof of Theorem 1

**Over-parameterized Regime**

We first provide the proof for the 1-layer model. The values of $\mathcal{L}\left(\theta^\star\right), \mathcal{L}\left(\theta^\star + \Delta\theta\right)$ are provided as

$$\mathcal{L}\left(\theta^\star\right) = \frac{1}{m}\sum_{i\in\mathcal{S}}|\varepsilon_i|, \quad \mathcal{L}\left(\theta^\star + \Delta\theta\right) = \frac{1}{m}\sum_{i\in\bar{\mathcal{S}}}|\langle x_i, \Delta\theta\rangle| + \frac{1}{m}\sum_{i\in\mathcal{S}}|\langle x_i, \Delta\theta\rangle - \varepsilon_i|. \quad (14)$$

Here $\mathcal{S} \in [m] : \{1, 2, \ldots, m\}$ is the support of the noise vector $\varepsilon = [\varepsilon_1, \cdots, \varepsilon_m]^\top$, and $\bar{\mathcal{S}} = [m] - \mathcal{S}$. Hence, we have

$$\mathcal{L}\left(\theta^\star + \Delta\theta\right) - \mathcal{L}\left(\theta^\star\right) = \frac{1}{m}\sum_{i\in\mathcal{S}}\left(|\langle x_i, \Delta\theta\rangle - \varepsilon_i| - |\varepsilon_i|\right) + \frac{1}{m}\sum_{i\in\bar{\mathcal{S}}}|\langle x_i, \Delta\theta\rangle|. \quad (15)$$

For simplicity, we denote $\mathbf{X} = [x_1, \cdots, x_m]^\top \in \mathbb{R}^{m\times d}$ by the design matrix. To upper bound the infimum of $\mathcal{L}\left(\theta^\star + \Delta\theta\right) - \mathcal{L}\left(\theta^\star\right)$ over the $\ell_\infty$-norm ball $\{\Delta\theta : \|\Delta\theta\|_\infty \leq \gamma\}$, we choose a specific $\overline{\Delta\theta}$ for each realization $\{(x_i, y_i)\}_{i=1}^m$ and show that $\overline{\Delta\theta}$ lies in the $\ell_\infty$-norm ball with high probability. To this goal, we pick $\overline{\Delta\theta}$ by solving the following linear equation

$$\mathbf{X}\Delta\theta = b, \quad (16)$$

where $b[i] = \text{Sign}(\varepsilon_i)\left(\xi \wedge |\varepsilon_i|\right)$ for $i \in \mathcal{S}$, and $b[i] = 0$ otherwise. Here $\xi > 0$ is a hyperparameter to be tuned later. Before showing that such $\overline{\Delta\theta}$ lies in the $\ell_\infty$-norm ball with high probability, we first further upper bound $\mathcal{L}\left(\theta^\star + \Delta\theta\right) - \mathcal{L}\left(\theta^\star\right)$ for this specific choice of $\overline{\Delta\theta}$. By our construction of $\overline{\Delta\theta}$, we first have

$$\mathcal{L}\left(\theta^\star + \overline{\Delta\theta}\right) - \mathcal{L}\left(\theta^\star\right) = \frac{1}{m}\sum_{i\in\mathcal{S}}\left(|\langle x_i, \overline{\Delta\theta}\rangle - \varepsilon_i| - |\varepsilon_i|\right) + \frac{1}{m}\sum_{i\in\bar{\mathcal{S}}}|\langle x_i, \overline{\Delta\theta}\rangle|$$

$$= -\frac{1}{m}\sum_{i\in\mathcal{S}}\xi \wedge |\varepsilon_i|. \quad (17)$$

On the other hand, based on Assumption 1, each $\varepsilon_i, i \in \mathcal{S}$ has nonzero probability to be away from zero. Hence, we can define the event $\mathcal{E} := \{$There are at least $p_0 pm$ elements of $|\varepsilon|$ larger than $t_0\}$. Define $\mathcal{M}_0 = \{i : |\varepsilon_i| \geq t_0\}$. Using the tail bound of binomial distribution, we have $\mathbb{P}(\mathcal{E}) \geq \frac{1}{4}$. Conditioned on the event $\mathcal{E}$, we have $m_0 = |\mathcal{M}_0| \geq p_0 pm$ and

$$\mathcal{L}\left(\theta^\star + \overline{\Delta\theta}\right) - \mathcal{L}\left(\theta^\star\right) = -\frac{1}{m} \sum_{i \in \mathcal{S}} \xi \wedge |\varepsilon_i| \leq -p_0 p \cdot (\xi \wedge t_0). \tag{18}$$

Now it suffices to show that such $\overline{\Delta\theta}$ belongs to the $\ell_\infty$-norm ball with high probability. To this goal, we first choose a specific $\overline{\Delta\theta}$ satisfying $\mathbf{X}\overline{\Delta\theta} = b$. We divide $\mathbf{X} = [\mathbf{X}_1, \mathbf{X}_2]$ where $\mathbf{X}_1 \in \mathbb{R}^{m \times m}$ and $\mathbf{X}_2 \in \mathbb{R}^{m \times (d-m)}$. Note that $\mathbf{X}_1$ is non-singular almost surely. Hence, we can pick $\overline{\Delta\theta} = [b^\top \mathbf{X}_1 (\mathbf{X}_1^\top \mathbf{X}_1)^{-1}, \mathbf{0}]^\top$. Next, it suffices to verify that with high probability, $\left\|\overline{\Delta\theta}\right\|_\infty \leq \gamma$, which is equivalent to

$$\left\|(\mathbf{X}_1^\top \mathbf{X}_1)^{-1} \mathbf{X}_1^\top b\right\|_\infty \leq \gamma. \tag{19}$$

To this goal, note that with probability at least $1 - e^{-\Omega(m)}$

$$\begin{aligned}
\left\|(\mathbf{X}_1^\top \mathbf{X}_1)^{-1} \mathbf{X}_1^\top b\right\|_\infty &\leq \left\|(\mathbf{X}_1^\top \mathbf{X}_1)^{-1} \mathbf{X}_1^\top b\right\| \\
&\leq \left\|\left(\frac{1}{m}\mathbf{X}_1^\top \mathbf{X}_1\right)^{-1}\right\| \left\|\frac{1}{m}\mathbf{X}_1\right\| \|b\| \\
&\overset{(a)}{\lesssim} \frac{1}{\sqrt{m}} \|b\| \\
&\lesssim \xi.
\end{aligned} \tag{20}$$

Here in (a) we used the fact that $\left\|\left(\frac{1}{m}\mathbf{X}_1^\top \mathbf{X}_1\right)^{-1}\right\| \lesssim 1$ and $\left\|\frac{1}{m}\mathbf{X}_1\right\| \lesssim \frac{1}{\sqrt{m}}$ with probability at least $1 - e^{-\Omega(m)}$. Hence, we only need to set $\xi \lesssim \gamma$, so that the chosen $\overline{\Delta\theta}$ belongs to the $\ell_\infty$-norm ball with probability at least $1 - e^{-\Omega(m)}$.

Combining the above steps and choosing $\xi \lesssim \gamma$, we know that with probability at least $\frac{1}{16}$, the below upper bound holds

$$\inf_{\|\Delta\theta\|_\infty \leq \gamma} \{\mathcal{L}(\theta^\star + \Delta\theta) - \mathcal{L}(\theta^\star)\} \overset{\text{w.p. } 1-e^{-\Omega(m)}}{\leq} \mathcal{L}(\theta^\star + \overline{\Delta\theta}) - \mathcal{L}(\theta^\star)$$
$$\overset{\text{w.p. } \frac{1}{4}}{\leq} -p_0 p\xi \tag{21}$$
$$\lesssim -p_0 p\gamma.$$

This completes the proof for the 1-layer model. For general $N$-layer models, we consider the true solution $\mathbf{w} \in \mathcal{W}$ with $w_1 = \theta^\star$, and $w_2 = \cdots = w_N = \mathbf{1}$. Moreover, for $\Delta\mathbf{w}$ we choose $\Delta w_2 = \cdots \Delta w_N = \mathbf{0}$. It is easy to verify that

$$\mathcal{L}(\mathbf{w}) - \mathcal{L}(\mathbf{w} + \Delta\mathbf{w}) = \frac{1}{m} \sum_{i \in \mathcal{S}} (|\langle x_i, \Delta w_1\rangle - \varepsilon_i| - |\varepsilon_i|) + \frac{1}{m} \sum_{i \in \bar{\mathcal{S}}} |\langle x_i, \Delta w_1\rangle|. \tag{22}$$

Therefore, an argument similar to the 1-layer model can be used to write

$$\inf_{\mathbf{w} \in \mathcal{W}} \inf_{\mathbf{w}': \|\mathbf{w}-\mathbf{w}'\|_\infty \leq \gamma} \{\mathcal{L}(\mathbf{w}) - \mathcal{L}(\mathbf{w}')\} \leq \inf_{\|\Delta w_1\|_\infty \leq \gamma} \{\mathcal{L}(\mathbf{w}) - \mathcal{L}(\mathbf{w} + \Delta\mathbf{w})\} \lesssim -p_0 p\gamma, \tag{23}$$

with probability of $\frac{1}{16}$.

**Under-parameterized Regime**

Given $\mathbf{w} \in \mathcal{W}$ and any $\Delta\mathbf{w}$, consider $\mathbf{w}' = \mathbf{w} + \Delta\mathbf{w}$ and define

$$\Delta\theta = \sum_{i=1}^{N} (\theta^\star)^{\frac{N-i}{N}} \odot \sum_{j_1, \cdots, j_i} \Delta w_{j_1} \odot \cdots \odot \Delta w_{j_i}. \tag{24}$$

We have

$$\mathcal{L}(\mathbf{w}) - \mathcal{L}(\mathbf{w} + \Delta\mathbf{w}) = \frac{1}{m}\sum_{i\in\mathcal{S}}\left(|\langle x_i, \Delta\theta\rangle - \varepsilon_i| - |\varepsilon_i|\right) + \frac{1}{m}\sum_{i\in\bar{\mathcal{S}}}|\langle x_i, \Delta\theta\rangle|$$

$$\geq \frac{1}{m}\sum_{i\in\bar{\mathcal{S}}}|\langle x_i, \Delta\theta\rangle| - \frac{1}{m}\sum_{i\in\mathcal{S}}|\langle x_i, \Delta\theta\rangle|. \tag{25}$$

Hence, it suffices to show that, with probability of $1 - e^{-\Omega(d)}$,

$$\inf_{\Delta\theta\in\mathbb{R}^d}\left\{\frac{1}{m}\sum_{i\in\bar{\mathcal{S}}}|\langle x_i, \Delta\theta\rangle| - \frac{1}{m}\sum_{i\in\mathcal{S}}|\langle x_i, \Delta\theta\rangle|\right\} \geq 0. \tag{26}$$

Note that the above inequality is invariant with respect to scaling. Hence, it suffices to show that it holds for arbitrary $\Delta\theta \in \mathbb{S}^{d-1}$ where $\mathbb{S}^{d-1} := \{x \in \mathbb{R}^d : \|x\| = 1\}$ is the standard sphere. Hence, it suffices to show

$$\inf_{\Delta\theta\in\mathbb{S}^{d-1}}\left\{\frac{1}{m}\sum_{i\in\bar{\mathcal{S}}}|\langle x_i, \Delta\theta\rangle| - \frac{1}{m}\sum_{i\in\mathcal{S}}|\langle x_i, \Delta\theta\rangle|\right\}$$

$$\geq \inf_{\Delta\theta\in\mathbb{S}^{d-1}}\left\{\frac{1}{m}\sum_{i\in\bar{\mathcal{S}}}|\langle x_i, \Delta\theta\rangle|\right\} - \sup_{\Delta\theta\in\mathbb{S}^{d-1}}\left\{\frac{1}{m}\sum_{i\in\mathcal{S}}|\langle x_i, \Delta\theta\rangle|\right\} \geq 0. \tag{27}$$

For the first term, applying Lemma 6, we have that with probability at least $1 - e^{-\Omega(d)}$

$$\inf_{\Delta\theta\in\mathbb{S}^{d-1}}\frac{1}{m}\sum_{i\in\bar{\mathcal{S}}}|\langle x_i, \Delta\theta\rangle| \geq \sqrt{\frac{2}{\pi}}(1-p) - \sqrt{\frac{(1-p)d}{m}}. \tag{28}$$

Similarly, for the second part, with probability of at least $1 - e^{-\Omega(d)}$, we have

$$\sup_{\Delta\theta\in\mathbb{S}^{d-1}}\frac{1}{m}\sum_{i\in\mathcal{S}}|\langle x_i, \Delta\theta\rangle| \leq \sqrt{\frac{2}{\pi}}p + \sqrt{\frac{pd}{m}}. \tag{29}$$

Combining both parts, we have that with probability at least $1 - e^{-\Omega(d)}$

$$\inf_{\Delta\theta\in\mathbb{S}^{d-1}}\left\{\frac{1}{m}\sum_{i\in\bar{\mathcal{S}}}|\langle x_i, \Delta\theta\rangle|\right\} - \sup_{\Delta\theta\in\mathbb{S}^{d-1}}\left\{\frac{1}{m}\sum_{i\in\mathcal{S}}|\langle x_i, \Delta\theta\rangle|\right\} \geq \sqrt{\frac{2}{\pi}}(1-2p) - 2\sqrt{\frac{d}{m}} \geq 0. \tag{30}$$

The last inequality follows from the fact that $m \gtrsim \frac{d}{(1-2p)^2}$. This completes the proof. $\square$

## C.2 Proof of Theorem 4

Given any $\Delta\mathbf{w} = [\Delta w_1, \cdots, \Delta w_N]^\top$, the following equality holds for any point $\mathbf{w} = \mathbf{w}^\star + \Delta\mathbf{w}$ where $\mathbf{w}^\star = [\sqrt[N]{\theta^\star}, \cdots, \sqrt[N]{\theta^\star}]^\top$:

$$(\sqrt[N]{\theta^\star} + \Delta w_1) \odot \cdots \odot (\sqrt[N]{\theta^\star} + \Delta w_N) - \theta^\star = \underbrace{\sum_{i=1}^{N-1}(\theta^\star)^{\frac{N-i}{N}} \odot \sum_{j_1,\cdots,j_i}\Delta w_{j_1} \odot \cdots \odot \Delta w_{j_i}}_{:=\Delta\theta_1}$$

$$+ \underbrace{\Delta w_1 \odot \cdots \odot \Delta w_N}_{:=\Delta\theta_2}. \tag{31}$$

Hence, we have

$$\mathcal{L}(\mathbf{w}^\star) - \mathcal{L}(\mathbf{w}^\star + \Delta\mathbf{w}) = \frac{1}{m}\sum_{i\in\mathcal{S}}\left(|\langle x_i, \Delta\theta_1 + \Delta\theta_2\rangle - \varepsilon_i| - |\varepsilon_i|\right) + \frac{1}{m}\sum_{i\in\bar{\mathcal{S}}}|\langle x_i, \Delta\theta_1 + \Delta\theta_2\rangle|. \tag{32}$$

For simplicity, we denote $\Theta_1$ as the set of $\Delta\theta_1$ defined in (31) with $\|\Delta\mathbf{w}\|_\infty \leq \gamma$. Similarly, $\Theta_2$ is the set of $\Delta\theta_2$ defined in (31) with $\|\Delta\mathbf{w}\|_\infty \leq \gamma$.

**Lower bound.** To prove the lower bound, one can write

$$\mathcal{L}(\mathbf{w}^\star) - \mathcal{L}(\mathbf{w}^\star + \Delta\mathbf{w}) \geq \frac{1}{m}\sum_{i\in\bar{\mathcal{S}}} |\langle x_i, \Delta\theta_1 + \Delta\theta_2\rangle| - \frac{1}{m}\sum_{i\in\mathcal{S}} |\langle x_i, \Delta\theta_1 + \Delta\theta_2\rangle|$$

$$\geq \frac{1}{m}\sum_{i\in\bar{\mathcal{S}}} |\langle x_i, \Delta\theta_1\rangle| - \frac{1}{m}\sum_{i\in\mathcal{S}} |\langle x_i, \Delta\theta_1\rangle| - \frac{1}{m}\sum_{i=1}^{m} |\langle x_i, \Delta\theta_2\rangle|$$

(33)

Hence, we have

$$\inf_{\|\Delta\mathbf{w}\|_\infty \leq \gamma} \{\mathcal{L}(\mathbf{w}^\star) - \mathcal{L}(\mathbf{w}^\star + \Delta\mathbf{w})\} \geq \inf_{\Delta\theta_1\in\Theta_1} \left\{ \frac{1}{m}\sum_{i\in\bar{\mathcal{S}}} |\langle x_i, \Delta\theta_1\rangle| - \frac{1}{m}\sum_{i\in\mathcal{S}} |\langle x_i, \Delta\theta_1\rangle| \right\}$$

$$- \sup_{\Delta\theta_2\in\Theta_2} \left\{ \frac{1}{m}\sum_{i=1}^{m} |\langle x_i, \Delta\theta_2\rangle| \right\}.$$

(34)

First, we bound the term $\inf_{\Delta\theta_1\in\Theta_1} \left\{ \frac{1}{m}\sum_{i\in\bar{\mathcal{S}}} |\langle x_i, \Delta\theta_1\rangle| - \frac{1}{m}\sum_{i\in\mathcal{S}} |\langle x_i, \Delta\theta_1\rangle| \right\}$. It is easy to see that the vector $\Delta\theta_1$ is $k$-sparse and has the same sparsity pattern as $\theta^\star$. Therefore, according to Lemma 6, there exist universal constants $C, c > 0$ such that the following hold

$$\mathbb{P}\left( \sup_{\Delta\theta_1\in\Theta_1} \left| \frac{1}{m'\|\Delta\theta_1\|}\sum_{i\in\bar{\mathcal{S}}} |\langle x_i, \Delta\theta_1\rangle| - \sqrt{\frac{2}{\pi}} \right| \geq C\sqrt{\frac{k}{m'}} + \delta \right) \leq e^{-cm'\delta^2},$$

(35)

and

$$\mathbb{P}\left( \sup_{\Delta\theta_1\in\Theta_1} \left| \frac{1}{m''\|\Delta\theta_1\|}\sum_{i\in\mathcal{S}} |\langle x_i, \Delta\theta_1\rangle| - \sqrt{\frac{2}{\pi}} \right| \geq C\sqrt{\frac{k}{m''}} + \delta \right) \leq e^{-cm''\delta^2}.$$

(36)

Here $m' = (1-p)m$, and $m'' = pm$. The inequality (35) implies that

$$\mathbb{P}\left( \frac{1}{m}\sum_{i\in\bar{\mathcal{S}}} |\langle x_i, \Delta\theta_1\rangle| \geq \|\Delta\theta_1\| \left( \sqrt{\frac{2}{\pi}}(1-p) - C\sqrt{\frac{(1-p)k}{m}} - \delta_1 \right), \forall\Delta\theta_1\in\Theta_1 \right) \geq 1 - e^{-\frac{cm\delta_1^2}{1-p}}.$$

(37)

Similarly, the inequality (36) leads to

$$\mathbb{P}\left( \frac{1}{m}\sum_{i\in\mathcal{S}} |\langle x_i, \Delta\theta_1\rangle| \leq \|\Delta\theta_1\| \left( \sqrt{\frac{2}{\pi}}p + C\sqrt{\frac{pk}{m}} + \delta_2 \right), \forall\Delta\theta_1\in\Theta_1 \right) \geq 1 - e^{-\frac{cm\delta_2^2}{p}}. \quad (38)$$

Upon setting $\delta_1 = \sqrt{\frac{(1-p)k}{m}}$ and $\delta_2 = \sqrt{\frac{pk}{m}}$, with probability of $1 - e^{-\Omega(k)}$, for all $\Delta\theta_1 \in \Theta_1$, we have

$$\frac{1}{m}\sum_{i\in\bar{\mathcal{S}}} |\langle x_i, \Delta\theta_1\rangle| - \frac{1}{m}\sum_{i\in\mathcal{S}} |\langle x_i, \Delta\theta_1\rangle| \geq \sqrt{\frac{2}{\pi}}(1-2p) - C'\sqrt{\frac{k}{m}} \geq 0. \quad (39)$$

In the last inequality we used the assumption $m \gtrsim \frac{k}{(1-2p)^2}$. The above argument implies

$$\inf_{\Delta\theta_1\in\Theta_1} \left\{ \frac{1}{m}\sum_{i\in\bar{\mathcal{S}}} |\langle x_i, \Delta\theta_1\rangle| - \frac{1}{m}\sum_{i\in\mathcal{S}} |\langle x_i, \Delta\theta_1\rangle| \right\} \geq 0, \quad (40)$$

with probability of at least $1 - e^{-\Omega(k)}$. Now we turn to bound the second part $\sup_{\Delta\theta_2\in\Theta_2} \left\{ \frac{1}{m}\sum_{i=1}^{m} |\langle x_i, \Delta\theta_2\rangle| \right\}$. To this goal, we first apply Lemma 6, which leads to

$$\mathbb{P}\left( \sup_{\Delta\theta_2\in\Theta_2} \left| \frac{1}{m\|\Delta\theta_2\|}\sum_{i=1}^{m} |\langle x_i, \Delta\theta_2\rangle| - \sqrt{\frac{2}{\pi}} \right| \geq C\sqrt{\frac{d}{m}} + \delta \right) \leq e^{-cm\delta^2}. \quad (41)$$

Therefore, upon setting $\delta = \sqrt{\frac{d}{m}}$, with probability of $1 - e^{-\Omega(d)}$, we have

$$
\begin{aligned}
\sup_{\Delta\theta_2 \in \Theta_2} \left\{ \frac{1}{m} \sum_{i=1}^{m} |\langle x_i, \Delta\theta_2 \rangle| \right\} &\leq \sup_{\Delta\theta_2 \in \Theta_2} \|\Delta\theta_2\| \left( \sqrt{\frac{2}{\pi}} + (C+1)\sqrt{\frac{d}{m}} \right) \\
&\lesssim \sqrt{\frac{d}{m}} \sup_{\|\mathbf{w}\|_\infty \leq \gamma} \|\Delta w_1 \odot \cdots \odot \Delta w_N\| \\
&= \sqrt{\frac{d}{m}} \sup_{|\Delta w_i[j]| \leq \gamma} \sqrt{\sum_{j \in [d]} \left( \prod_{i \in [N]} \Delta w_i[j] \right)^2} \\
&= \frac{d}{\sqrt{m}} \gamma^N.
\end{aligned}
\tag{42}
$$

Here $\Delta w_i[j]$ is the $j$-th element of $\Delta w_i$. Therefore, we conclude that

$$
\inf_{\|\Delta\mathbf{w}\|_\infty \leq \gamma} \{ \mathcal{L}(\mathbf{w}^\star) - \mathcal{L}(\mathbf{w}^\star + \Delta\mathbf{w}) \} \gtrsim -\frac{d}{\sqrt{m}} \gamma^N,
\tag{43}
$$

with probability of $1 - e^{-\Omega(k)}$, thereby completing the proof of the lower bound.

**Upper bound.** To this goal, we first define a restricted set of perturbation vectors

$$
\mathcal{V} := \{ \mathbf{v} : \|\mathbf{v}\|_\infty \leq \gamma, v_i[j] = 0, \forall i \in [N], j \in \mathrm{supp}(\theta^\star) \},
\tag{44}
$$

where $\mathrm{supp}(\theta^\star)$ is the support of $\theta^\star$. Based on this definition, we have

$$
\begin{aligned}
\inf_{\|\Delta\mathbf{w}\|_\infty \leq \gamma} \{ \mathcal{L}(\mathbf{w}^\star) - \mathcal{L}(\mathbf{w}^\star + \Delta\mathbf{w}) \} &\leq \inf_{\Delta\mathbf{w} \in \mathcal{V}} \{ \mathcal{L}(\mathbf{w}^\star) - \mathcal{L}(\mathbf{w}^\star + \Delta\mathbf{w}) \} \\
&= \inf_{\Delta\mathbf{w} \in \mathcal{V}} \left\{ \frac{1}{m} \sum_{i \in \mathcal{S}} (|\langle x_i, \Delta\theta_2 \rangle - \varepsilon_i| - |\varepsilon_i|) + \frac{1}{m} \sum_{i \in \bar{\mathcal{S}}} |\langle x_i, \Delta\theta_2 \rangle| \right\}.
\end{aligned}
\tag{45}
$$

where $\Delta\theta_2 = \Delta w_1 \odot \cdots \odot \Delta w_N$ is the same as before. Note that for any $\|\Delta\mathbf{w}\|_\infty \leq \gamma$, we have $\|\Delta\theta_2\| \leq \sqrt{d}\gamma^N$. Moreover, this bound is attainable when $\Delta w_i \equiv [\pm\gamma, \cdots, \pm\gamma]^\top$. Hence, we have

$$
\begin{aligned}
&\inf_{\Delta\mathbf{w} \in \mathcal{V}} \left\{ \frac{1}{m} \sum_{i \in \mathcal{S}} (|\langle x_i, \Delta\theta_2 \rangle - \varepsilon_i| - |\varepsilon_i|) + \frac{1}{m} \sum_{i \in \bar{\mathcal{S}}} |\langle x_i, \Delta\theta_2 \rangle| \right\} \\
&\leq \inf_{\|\Delta\theta_2\| \leq \sqrt{d}\gamma^N} \left\{ \frac{1}{m} \sum_{i \in \mathcal{S}} (|\langle x_i, \Delta\theta_2 \rangle - \varepsilon_i| - |\varepsilon_i|) + \frac{1}{m} \sum_{i \in \bar{\mathcal{S}}} |\langle x_i, \Delta\theta_2 \rangle| \right\}.
\end{aligned}
\tag{46}
$$

An argument similar to the proof of Theorem 1 can be used to show that, with probability of at least $1/16$, we have

$$
\inf_{\|\Delta\theta_2\| \leq \sqrt{d}\gamma^N} \left\{ \frac{1}{m} \sum_{i \in S} (|\langle x_i, \Delta\theta_2 \rangle - \varepsilon_i| - |\varepsilon_i|) + \frac{1}{m} \sum_{i \in \bar{S}} |\langle x_i, \Delta\theta_2 \rangle| \right\} \lesssim -\sqrt{\frac{p_0 p(d-k)}{m}} \sqrt{d}\gamma^N.
\tag{47}
$$

Recalling that $k \ll d$, we have with probability of $1/16$

$$
\inf_{\|\Delta\mathbf{w}\|_\infty \leq \gamma} \{ \mathcal{L}(\mathbf{w}^\star) - \mathcal{L}(\mathbf{w}^\star + \Delta\mathbf{w}) \} \lesssim -\sqrt{p_0 p} \frac{d}{\sqrt{m}} \gamma^N.
\tag{48}
$$

This completes the proof. $\qquad\square$

# D   Proofs of Convergence Analysis

## D.1   Proof of Theorem 2

For simplicity of notation, we denote $u = w_1$ and $v = w_2$. Moreover, without loss of generality, we assume that the elements of $\theta^\star$ are arranged in descending order, i.e., $\theta_1^\star \geq \cdots \geq \theta_k^\star > \theta_{k+1}^\star = \cdots = \theta_d^\star = 0$, and the initial point satisfies $u_i, v_i = \Theta(\sqrt{\alpha}), \forall i \in [d]$. Moreover, for $v \in \mathbb{R}^d$, we define $v_{:i} = [v_1, \cdots, v_i]^\top$ and $v_{i:} = [v_i, \cdots, v_d]^\top$. For short, we denote $v_{-i} = v_{i+1:}$. Finally, we define $\kappa = \theta_1^\star / \theta_k^\star$ as the condition number. Moreover, without loss of generality, we assume that $\theta_1^\star \geq 1 \geq \theta_k^\star$.

First, according to Proposition 1, the sub-differential of $\mathcal{L}(u, v)$ is uniformly concentrated around its population gradient. In particular, with probability at least $1 - Ce^{-cm\delta^2}$, we have

$$u_{i,t+1} = u_{i,t} + \eta \frac{\theta_i^\star - u_{i,t}v_{i,t}}{\|u_t \odot v_t - \theta^\star\|} v_{i,t} + \eta \delta_i v_{i,t}, \text{ and } |\delta_i| \leq \delta, \forall i \in [d], \tag{49}$$

$$v_{i,t+1} = v_{i,t} + \eta \frac{\theta_i^\star - u_{i,t}v_{i,t}}{\|u_t \odot v_t - \theta^\star\|} u_{i,t} + \eta \delta_i u_{i,t}, \text{ and } |\delta_i| \leq \delta, \forall i \in [d]. \tag{50}$$

Hence, we have

$$u_{i,t+1}v_{i,t+1} = u_{i,t}v_{i,t} + \eta \left( \frac{\theta_i^\star - u_{i,t}v_{i,t}}{\|u_t \odot v_t - \theta^\star\|} + \delta_i \right) (u_{i,t}^2 + v_{i,t}^2) + \eta^2 \left( \frac{\theta_i^\star - u_{i,t}v_{i,t}}{\|u_t \odot v_t - \theta^\star\|} + \delta_i \right)^2 u_{i,t}v_{i,t}. \tag{51}$$

Moreover, we have

$$u_{i,t+1}^2 + v_{i,t+1}^2 = (u_{i,t}^2 + v_{i,t}^2) \left( 1 + \eta^2 \left( \frac{(\theta_i^\star - u_{i,t}v_{i,t})}{\|u_t \odot v_t - \theta^\star\|} + \delta_i \right)^2 \right) + 4\eta \left( \frac{\theta_i^\star - u_{i,t}v_{i,t}}{\|u_t \odot v_t - \theta^\star\|} + \delta_i \right) u_{i,t}v_{i,t}. \tag{52}$$

**Signal Dynamics**

We first study the behavior of the signal term $S_t = u_{:k,t}v_{:k,t}$ for the first $k$ components of the model $u_t \odot v_t$. We divide the dynamics into $k + 1$ stages. In the first $k$ stages, each component $u_{i,t}v_{i,t}$ converges to $\theta_i^\star$ sequentially. Once all the components are close to the ground truth, the distance between signal term and the ground truth $\|S_t - \theta_{:k}^\star\|$ will further decrease to $\mathcal{O}\left(\sqrt{d^2m}\alpha^2 \vee \eta\theta_1^\star\right)$.

**Stage 1:**   In this stage, the first component $u_1v_1$ grows to $\theta_1 - \delta\|\theta^\star\|$ within $\Theta\left(\frac{\|\theta^\star\|}{\eta\theta_1^\star} \log\left(\frac{1}{\alpha}\right)\right)$ iterations. At the initial point, we have $u_{1,0}v_{1,0} = \Theta(\alpha)$. For iteration $t + 1$, according to (51), we have

$$u_{1,t+1}v_{1,t+1} \geq u_{1,t}v_{1,t} + \eta \left( \frac{\theta_1^\star - u_{1,t}v_{1,t}}{\|u_t \odot v_t - \theta^\star\|} + \delta_1 \right) (u_{1,t}^2 + v_{1,t}^2)$$
$$\geq u_{1,t}v_{1,t} + 2\eta \left( \frac{\theta_1^\star - u_{1,t}v_{1,t}}{\|u_t \odot v_t - \theta^\star\|} + \delta_1 \right) u_{1,t}v_{1,t}. \tag{53}$$

We further divide our analysis into two substages. In the first substage, we have $u_1v_1 \leq \theta_1^\star / 2$. Note that $\|u_t \odot v_t - \theta^\star\| = \mathcal{O}(\|\theta^\star\|)$ and $|\delta_1| \leq \delta \lesssim \frac{1}{\kappa}$. Hence, (53) can be further simplified as

$$u_{1,t+1}v_{1,t+1} \geq \left( 1 + \frac{1}{4} \frac{\eta\theta_1^\star}{\|\theta^\star\|} \right) u_{1,t}v_{1,t}. \tag{54}$$

Therefore, this stage ends within $\mathcal{O}\left(\frac{\|\theta^\star\|}{\eta\theta_1^\star} \log\left(\frac{1}{\alpha}\right)\right)$ iterations. In the second substage, we have $u_1v_1 \geq \theta_1^\star / 2$. Upon defining $x_t = \theta_1^\star - u_{1,t}v_{1,t}$, one can write

$$x_{t+1} \leq \left( 1 - 2\eta \frac{u_{1,t}v_{1,t}}{\|u_t \odot v_t - \theta^\star\|} \right) x_t - 2\eta\delta_1 u_t \odot v_t$$
$$\leq \left( 1 - \frac{\eta\theta_1^\star}{\|\theta^\star\|} \right) x_t + \eta\delta\theta_1^\star. \tag{55}$$

Hence, within additional $\mathcal{O}\left(\|\theta^\star\|/(\eta\theta_1^\star)\right)$ iterations, we have $u_{1,T_1}v_{1,T_1} = \theta_1^\star \pm \delta\|\theta^\star\|$. Overall, within $T_1 = \mathcal{O}\left(\frac{\|\theta^\star\|}{\eta\theta_1^\star}\log\left(\frac{1}{\alpha}\right)\right)$ iterations, we have $u_{1,T_1}v_{1,T_1} = \theta_1^\star \pm \delta\|\theta^\star\|$. Now, we turn to show the lower bound on $T_1$ by analyzing the trajectory of $u_{1,t}^2 + v_{1,t}^2$. Due to (52), when $u_{1,t}^2 + v_{1,t}^2 \leq \frac{\theta_1^\star}{2}$, we have

$$u_{1,t+1}^2 + v_{1,t+1}^2 \leq \left(u_{1,t}^2 + v_{1,t}^2\right)\left(1 + 10\frac{\eta\theta_1^\star}{\|\theta^\star\|}\right). \tag{56}$$

Hence, at least $\Omega\left(\frac{\|\theta^\star\|}{\eta\theta_1^\star}\log\left(\frac{1}{\alpha}\right)\right)$ iterations are needed for $u_{1,t}^2 + v_{1,t}^2$ to be larger than $\frac{\theta_1^\star}{2}$. Since $u_{1,t}^2 + v_{1,t}^2 \geq u_{1,t}v_{1,t}$, we immediately obtain $T_1 = \Omega\left(\frac{\|\theta^\star\|}{\eta\theta_1^\star}\log\left(\frac{1}{\alpha}\right)\right)$.

**Stage 2 to Stage $k$:** In the next $k-1$ stages, each component $u_iv_i$ will converge to $\theta_i^\star \pm \delta\|\theta^\star\|$ sequentially. To show this, we use an inductive argument. In each stage $i$, we assume that the first $i-1$ components have already converged close to $\theta_j^\star, \forall j \in [i-1]$. Hence, we have $\|u_t \odot v_t - \theta^\star\| = \Theta\left(\left\|\theta_{-(i-1)}^\star\right\|\right)$. Repeating the procedure in Stage 1, we can show that, at stage $i$, $T_i = \mathcal{O}\left(\frac{\|\theta_{-(i-1)}^\star\|}{\eta\theta_i^\star}\log\left(\frac{1}{\alpha}\right)\right)$ iterations are needed for $u_iv_i$ to converge to $\theta_i^\star \pm \delta\|\theta^\star\|$. Overall, after $T = T_1 + \cdots T_k = \mathcal{O}\left(\frac{k^{3/2}}{\eta}\log\left(\frac{1}{\alpha}\right)\right)$ iterations, we have

$$\left\|u_{:k,T}v_{:k,T} - \theta_{:k,T}^\star\right\| \lesssim \sqrt{k}\delta\|\theta^\star\|. \tag{57}$$

**Stage $k+1$:** In the final stage, the signal term will quickly decrease to $\mathcal{O}\left(\sqrt{d^2m}\alpha^{1-\Theta(\delta)} \vee \eta\theta_1^\star\right)$ within $T_{k+1} = \mathcal{O}\left(\frac{\delta\|\theta^\star\|}{\eta\theta_k^\star}\right)$ iterations. To show this, we write

$$\theta_{:k}^\star - S_{t+1} = \theta_{:k}^\star - S_t - \eta\frac{u_{:k,t}^2 + v_{:k,t}^2}{\|u_t \odot v_t - \theta^\star\|}\odot(\theta_{:k}^\star - S_t) - \eta\boldsymbol{\delta}\left(u_t^2 + v_t^2\right) + \eta^2\left(\frac{\theta_{:k}^\star - S_t}{\|u_t \odot v_t - \theta^\star\|} + \boldsymbol{\delta}\right)^2\odot S_t. \tag{58}$$

Here we denote $\boldsymbol{\delta} = [\delta_1, \cdots, \delta_k]^\top$. Note that $\|u_t \odot v_t - \theta^\star\| \leq \|\theta_{:k}^\star - S_t\| + \|E_t\|$. Moreover, based on our assumption, we have $\|E_t\| \lesssim \sqrt{d}\alpha^{1-\Theta(\delta)}$. Finally, the balanced property implies that $|u_{i,t} - v_{i,t}| = \mathcal{O}\left(\alpha^{1-\Theta(\delta)}\right)$ (this will be proven later). Hence, we have

$$\begin{aligned}
\|\theta_{:k}^\star - S_{t+1}\| &\leq \left\|(\theta_{:k}^\star - S_t)\odot\left(\boldsymbol{1} - \eta\frac{u_t^2 + v_t^2}{\|u_t \odot v_t - \theta^\star\|} + \eta^2\frac{(\theta_{:k}^\star - S_t)\odot S_t}{\|u_t \odot v_t - \theta^\star\|^2}\right)\right\| + 4\eta\delta\|\theta^\star\| \\
&\leq \|\theta_{:k}^\star - S_t\|\left(1 - \frac{\eta}{2}\frac{\theta_k^\star}{\|\theta_{:k}^\star - S_t\| + \|E_t\|}\right) + 4\eta\delta\|\theta^\star\| \\
&\leq \|\theta_{:k}^\star - S_t\| - 0.25\eta\theta_k^\star + 4\eta\delta\|\theta^\star\| \\
&\leq \|\theta_{:k}^\star - S_t\| - 0.1\eta\theta_k^\star.
\end{aligned} \tag{59}$$

Here, we used the fact that $\delta \lesssim \frac{1}{\kappa}$. On the other hand, we have

$$\begin{aligned}
\|S_{t+1} - S_t\| &\leq \frac{\eta}{\|u_t \odot v_t - \theta^\star\|}\left\|\left(u_t^2 + v_t^2\right)\odot(\theta_{:k}^\star - S_t)\right\| + 4\eta\delta\|\theta^\star\| \\
&\overset{(a)}{\leq} 2\eta\theta_1^\star + 4\eta\delta\|\theta^\star\| \\
&\leq 3\eta\theta_1^\star,
\end{aligned} \tag{60}$$

where in (a) we used Lemma 7. The above inequality indicates that the signal propagation in each step is upper bounded by $\mathcal{O}\left(\eta\theta_1^\star\right)$. Hence, we conclude that within $T_{k+1} = \mathcal{O}\left(\frac{\delta\|\theta^\star\|}{\eta\theta_k^\star}\right)$ iterations, we have $\|\theta_{:k}^\star - S_t\| \lesssim \sqrt{d^2m}\alpha^{1-\Theta(\delta)} \vee \eta\theta_1^\star$. Since $\delta \lesssim \frac{1}{\kappa}$, the total iteration complexity is upper bounded by $T' = T_1 + \cdots T_{k+1} = \mathcal{O}\left(\frac{k^{3/2}}{\eta}\log\left(\frac{1}{\alpha}\right)\right)$.

**Residual Dynamics**

Now, we analyze the residual dynamics. Instead of analyzing the dynamics of $u_{i,t}v_{i,t}$, we analyze its surrogate $u_{i,t}^2 + v_{i,t}^2$. Based on (52), we can naturally bound it as follows

$$u_{i,t+1}^2 + v_{i,t+1}^2 \le u_{i,t}^2 + v_{i,t}^2 + 6\eta\delta u_{i,t}v_{i,t} \le (1 + 3\eta\delta)\left(u_{i,t}^2 + v_{i,t}^2\right). \tag{61}$$

Therefore, during the training process, we can bound the residual term as

$$u_{i,t}^2 + v_{i,t}^2 \lesssim \alpha\left(1 + \eta\delta\right)^{\mathcal{O}\left(\frac{k^{3/2}}{\eta}\log\left(\frac{1}{\alpha}\right)\right)} \lesssim \alpha^{1-\mathcal{O}\left(k^{3/2}\delta\right)}. \tag{62}$$

Hence, we have

$$\|E_t\| \le \left(\sum_{i=k+1}^{d} u_{i,t}^2 + v_{i,t}^2\right)^{1/2} \lesssim \sqrt{d}\alpha^{1-\mathcal{O}\left(k^{3/2}\delta\right)}. \tag{63}$$

Therefore, we conclude that within $\bar{T} = \mathcal{O}\left(\frac{k^{3/2}}{\eta}\log\left(\frac{1}{\alpha}\right)\right)$ iterations, we have

$$\|u_{\bar{T}} \odot v_{\bar{T}} - \theta^\star\| \le \|\theta_{:k}^\star - S_{\bar{T}}\| + \|E_{\bar{T}}\| \lesssim \sqrt{d^2m}\alpha^{1-\mathcal{O}(\delta)} \vee \eta\theta_1^\star. \tag{64}$$

Therefore, with probability at least $1 - e^{-C\log^2(m)\log(d)\log(\|\theta^\star\|/\alpha)}$, we have

$$\|u_{\bar{T}} \odot v_{\bar{T}} - \theta^\star\| \lesssim \sqrt{d^2m}\alpha^{1-\tilde{\Theta}\left(\frac{k^2}{\sqrt{m(1-p)^2}}\right)} \vee \eta\theta_1^\star. \tag{65}$$

**Long Escape Time**

Based on the above analysis, it can be seen that, after $\bar{T}$ iterations, the residual term is the dominant term, which may cause the algorithm to diverge, as captured by Figure 1. We now show that the residual term will not diverge within $T' = \sqrt{\frac{m(1-p)^2}{k}}\bar{T}$. To this goal, recall that for $\forall k + 1 \le i \le d$, we have

$$u_{i,t+1}^2 + v_{i,t+1}^2 \le (1 + 3\eta\delta)\left(u_{i,t}^2 + v_{i,t}^2\right). \tag{66}$$

Hence, for $t \le \frac{1}{6\eta\delta}\log\left(\frac{1}{\alpha}\right)$, we have

$$u_{i,t}^2 + v_{i,t}^2 \lesssim \alpha\left(1 + 3\eta\delta\right)^{\frac{1}{6\eta\delta}\log\left(\frac{1}{\alpha}\right)} \le \alpha e^{0.5\log\left(\frac{1}{\alpha}\right)} = \sqrt{\alpha}. \tag{67}$$

The proof is completed by noticing that $m = \tilde{\Omega}\left(\frac{k}{(1-p)^2\delta^2}\right)$.

**Balanced Property**

To prove the balanced property, we directly calculate the dynamic of the difference $u_{i,t} - v_{i,t}$. One can write

$$u_{i,t+1} - v_{i,t+1} = (u_{i,t} - v_{i,t})\left(1 - \eta\frac{\theta_i^\star - u_{i,t}v_{i,t}}{\|u_t \odot v_t - \theta^\star\|} - \eta\delta_i\right). \tag{68}$$

Since $\theta_i^\star - u_{i,t}v_{i,t} \ge 0$, we have

$$|u_{i,t+1} - v_{i,t+1}| \le |u_{i,t} - v_{i,t}|\left(1 + \eta\delta\right), \tag{69}$$

for $0 \le i \le k$. We conclude that

$$|u_{i,t} - v_{i,t}| \le \sqrt{\alpha}\left(1 + \eta\delta\right)^t \lesssim \alpha^{0.5-\Theta(k^{3/2}\delta)}. \tag{70}$$

for $\forall t \lesssim \frac{k^{3/2}}{\eta}\log\left(\frac{1}{\alpha}\right)$. On the other hand, for $i \ge k$, we can write

$$|u_{i,t} - v_{i,t}| \le \sqrt{u_{i,t}^2 + v_{i,t}^2} \lesssim \alpha^{0.5-\Theta(k^{3/2}\delta)}. \tag{71}$$

The proof is completed by noticing that $m = \tilde{\Omega}\left(\frac{k}{(1-p)^2\delta^2}\right)$.

**Convergence in Under-parameterized Regime**

In this section, we study the under-parameterized regime, where we assume that $m = \tilde{\Omega}(\frac{d}{(1-p)^2})$. The analysis of the signal term is the same as the over-parameterized regime and hence omitted for brevity. Here we only analyze the residual dynamic. One can write

$$E_{t+1} = E_t \left( 1 - \eta \frac{u_{k+1:,t}^2 + v_{k+1:,t}^2}{\|u_t \odot v_t - \theta^\star\|} \right) + \eta \delta_{k+1:} \left( u_{k+1:,t}^2 + v_{k+1:,t}^2 \right) + \eta^2 \left( \frac{-E_t}{\|u_t \odot v_t - \theta^\star\|} + \delta_{k+1:} \right)^2 E_t. \tag{72}$$

When the residual term becomes the dominant term, i.e., $\|\theta^\star_{:k} - S_t\| \leq \|E_t\|$, we have the simplified dynamic

$$
\begin{aligned}
\|E_{t+1}\| &\leq \left\| E_t \left( 1 - 0.5\eta \frac{u_{k+1:,t}^2 + v_{k+1:,t}^2}{\|u_t \odot v_t - \theta^\star\|} \right) + \eta \delta_{k+1:} \left( u_{k+1:,t}^2 + v_{k+1:,t}^2 \right) \right\| + 2\eta^2 \left( \|E_t\|^3 + \delta^2 \|E_t\| \right) \\
&\overset{(a)}{\leq} \left\| E_t \left( 1 - \frac{\eta E_t}{\|E_t\|} \right) \right\| + 4\eta\delta \|E_t\| \\
&\leq \left( 1 - \frac{\eta}{\sqrt{d}} \right) \|E_t\| + 4\eta\delta \|E_t\| \\
&\overset{(b)}{\leq} \left( 1 - 0.5\frac{\eta}{\sqrt{d}} \right) \|E_t\|.
\end{aligned}
\tag{73}
$$

Here in (a) we used the balanced property, which results in $u_{k+1:,t}^2 + v_{k+1:,t}^2 \asymp 2E_t$. Moreover, (b) is implied by the fact that $m \gtrsim \frac{d}{(1-p)^2}$, which in turn implies $\delta \lesssim \frac{1}{\sqrt{d}}$. Hence, we have

$$\|u_{t+1} \odot v_{t+1} - \theta^\star\| \leq \left( 1 - \Omega\left( \frac{\eta}{\sqrt{d}} \right) \right) \|u_t \odot v_t - \theta^\star\|. \tag{74}$$

Then, for $t \geq \bar{T}$, we have

$$\|u_t \odot v_t - \theta^\star\| \lesssim \sqrt{d^2 m} \alpha^{1 - \tilde{\Theta}\left( \frac{k^2}{\sqrt{m}(1-p)^2} \right)} \left( 1 - \Omega\left( \frac{\eta}{\sqrt{d}} \right) \right)^{t - \bar{T}} \vee \eta \theta^\star_1. \tag{75}$$

**D.2   Proof of Theorem 3**

The proof of $N$-layer model is similar to that of 2-layer model. First, we study the signal dynamics, showing that the first $k$ components $\prod w_{i,j}^{(t)}$ converge to $\theta^\star_j$ sequentially for $1 \leq j \leq k$. We also prove that the residual term remains small along the optimization trajectory. Based on the SubGM update rule, one can write

$$w_i^{(t+1)} = w_i^{(t)} + \eta \frac{\theta^\star - \prod w_j^{(t)}}{\left\| \theta^\star - \prod w_j^{(t)} \right\|} \prod_{j \neq i} w_j^{(t)} + \eta \delta \prod_{j \neq i} w_j^{(t)}, \text{ and } \|\delta\|_\infty \leq \delta, \forall i \in [N]. \tag{76}$$

**Signal Dynamics**

**Stage 1:** In this stage, we show that $\prod w_{i,1}$ will converge to $\theta^\star$ within $T_1 = \Theta\left(\frac{\|\theta^\star\|}{N\eta\theta_1^\star}\alpha^{-\frac{N-2}{N}}\right)$ iterations. We first prove the upper bound. According to the update rule, we have

$$
\begin{aligned}
\prod w_{i,1}^{(t+1)} &= \prod w_{i,1}^{(t)} + \sum_{i=1}^{N}\eta\left(\frac{\theta_1^\star - \prod w_{j,1}^{(t)}}{\left\|\theta^\star - \prod w_j^{(t)}\right\|} + \delta_1\right)\left(\prod_{j\neq i} w_{j,1}^{(t)}\right)^2 + \text{higher order terms of } \eta \\
&\geq \prod w_{i,1}^{(t)} + \sum_{i=1}^{N}\eta\left(\frac{\theta_1^\star - \prod w_{j,1}^{(t)}}{\left\|\theta^\star - \prod w_j^{(t)}\right\|} + \delta_1\right)\left(\prod_{j\neq i} w_{j,1}^{(t)}\right)^2 \\
&\geq \prod w_{i,1}^{(t)} + N\eta\left(\frac{\theta_1^\star - \prod w_{j,1}^{(t)}}{\left\|\theta^\star - \prod w_j^{(t)}\right\|} - \delta\right)\left(\prod w_{i,1}^{(t)}\right)^{\frac{2(N-1)}{N}}.
\end{aligned}
\tag{77}
$$

Here we use the fact that $\prod w_{i,1}^{(t)} \leq \theta_1^\star$ and $\eta \lesssim \frac{1}{N}\frac{1}{\kappa}^{\frac{N-2}{N}}$ so that we can drop the higher order terms. For brevity, we only show how we can drop the 2-th order term of $\eta$. The proof of the higher order terms is similar. One can write

$$
\begin{aligned}
\eta^2\sum_{i\neq j}\left(\prod_{k\neq i} w_{k,1}^{(t)}\prod_{k\neq j} w_{k,1}^{(t)}\prod_{k\neq i,j} w_{k,1}^{(t)}\right) &\overset{(a)}{\leq} \eta^2\left(\theta_1^\star\right)^{\frac{N-2}{N}}\sum_{i\neq j}\left(\prod_{k\neq i} w_{k,1}^{(t)}\prod_{k\neq j} w_{k,1}^{(t)}\right) \\
&\overset{(b)}{\leq} (N-1)\eta\left(\theta_1^\star\right)^{\frac{N-2}{N}}\eta\sum_{i=1}^{m}\left(\prod_{j\neq i} w_{j,1}^{(t)}\right)^2 \\
&\overset{(c)}{\lesssim} \eta\sum_{i=1}^{m}\left(\prod_{j\neq i} w_{j,1}^{(t)}\right)^2.
\end{aligned}
\tag{78}
$$

Here in (a) we use the balanced property and the fact that $\prod w_{i,1}^{(t)} \leq \theta_1^\star$. Moreover, in (b), we use the rearrangement Inequality. Finally in (c) we use the assumption that $\eta \lesssim N^{-1}\kappa^{-\frac{N-2}{N}}$. For simplicity, we denote $x_t = \prod w_{i,1}^{(t)}$. Note that $\left\|\theta^\star - \prod w_j^{(t)}\right\| \leq \|\theta^\star\|$. Hence, the dynamic can be simplified as

$$
x_{t+1} \geq x_t + \frac{N\eta}{\|\theta^\star\|}\left(\theta_1^\star - \delta\|\theta^\star\| - x_t\right)x_t^{\frac{2(N-1)}{N}}.
\tag{79}
$$

We next show that $x_T \geq \theta_1^\star - 2\delta\|\theta^\star\|$ within $T_1 = \Theta\left(\frac{\|\theta^\star\|}{N\eta\theta_1^\star}\alpha^{-\frac{N-2}{N}}\right)$ iterations provided that $x_0 = \Theta(\alpha)$. To this goal, we divide our analysis into two substages.

- $x_t \leq \frac{\theta_1^\star}{2}$: In this substage, we assume that $x_t \leq \frac{\theta_1^\star}{2}$. Hence, we can further simplify the dynamic as

$$
x_{t+1} \geq x_t + 0.5N\eta\frac{\theta_1^\star}{\|\theta^\star\|}x_t^{\frac{2(N-1)}{N}}.
\tag{80}
$$

Without loss of generality, we assume that $x_0 = \alpha$. Now we divide the interval $[\alpha, 0.5\theta_1^\star]$ into a series of sub-intervals $\{\mathcal{I}_k\}$, where $\mathcal{I}_k = [2^k\alpha, 2^{k+1}\alpha)$. In each $\mathcal{I}_k$, the dynamic can be further simplified as

$$
x_{t+1} \geq \left(1 + 0.5N\eta\frac{\theta_1^\star}{\|\theta^\star\|}\left(2^k\alpha\right)^{\frac{N-2}{N}}\right)x_t.
\tag{81}
$$

Therefore, the number of iterations that $x_t$ spends in each interval $\mathcal{I}_k$ is $\mathcal{O}\left(\frac{\|\theta^\star\|}{N\eta\theta_1^\star}\left(2^k\alpha\right)^{-\frac{N-2}{N}}\right)$. Hence, the total number of iterations is upper bounded by $\mathcal{O}\left(\sum_{k=0}^{\infty}\frac{\|\theta^\star\|}{N\eta\theta_1^\star}\left(2^k\alpha\right)^{-\frac{N-2}{N}}\right) = \mathcal{O}\left(\frac{\|\theta^\star\|}{N\eta\theta_1^\star}\alpha^{-\frac{N-2}{N}}\right)$.

- $x_t \geq \frac{\theta_1^\star}{2}$: In this substage, we define $y_t = \theta_1^\star - \delta \|\theta^\star\| - x_t$. Via a similar trick, we can show that within additional $\mathcal{O}\left( \frac{\|\theta^\star\|}{N\eta} (\theta_1^\star)^{-\frac{2N-2}{N}} \right)$ iterations, we have $x_t \geq \theta_1^\star - 2\delta \|\theta^\star\|$. Overall, after $T_1 = \Theta\left( \frac{\|\theta^\star\|}{N\eta\theta_1^\star} \alpha^{-\frac{N-2}{N}} \right)$ iterations, we have $\theta_1^\star - 2\delta \|\theta^\star\| \leq \prod w_{i,1}^{(T_1)} \leq \theta_1^\star$.

**Stages 2 to $k$:** Similarly, for component $\prod w_{j,i}^{(t)}$, it takes $\mathcal{O}\left( \frac{\|\theta_{-(i-1)}^\star\|}{N\eta\theta_i^\star} \alpha^{-\frac{N-2}{N}} \right)$ iterations to attain $\theta_i^\star - 2\delta \|\theta^\star\|$. Overall, Stages 2 to $k$ take $\Theta\left( \frac{k^{\frac{3}{2}}}{N\eta} \alpha^{-\frac{N-2}{N}} \right)$ iterations to terminate.

**Stage $k+1$:** In this stage, we take $S_t = \prod w_{j,:k}^{(t)}$. Hence, we have

$$
\begin{aligned}
\|\theta_{:k}^\star - S_{t+1}\| &\leq \left\| (\theta_{:k}^\star - S_t) \left( 1 - \eta \frac{\sum_{i=1}^N \left( \prod_{j\neq i} w_{j,1}^{(t)} \right)^2}{\left\| \theta^\star - \prod w_j^{(t)} \right\|} \right) \right\| + 4N\eta\delta\sqrt{k}(\theta_1^\star)^{\frac{2(N-1)}{N}} \\
&\leq \|\theta_{:k}^\star - S_t\| \left( 1 - N\eta \frac{(\theta_k^\star)^{\frac{2(N-1)}{N}}}{\|\theta_{:k}^\star - S_t\| + \|E_t\|} \right) + 4N\eta\delta\sqrt{k}(\theta_1^\star)^{\frac{2(N-1)}{N}} \\
&\leq \|\theta_{:k}^\star - S_t\| - 0.5N\eta(\theta_k^\star)^{\frac{2(N-1)}{N}} + 4N\eta\delta\sqrt{k}(\theta_1^\star)^{\frac{2(N-1)}{N}} \\
&\leq \|\theta_{:k}^\star - S_t\| - 0.1N\eta(\theta_k^\star)^{\frac{2(N-1)}{N}}.
\end{aligned}
\tag{82}
$$

Here we used the fact that $\left\| \theta^\star - \prod w_j^{(t)} \right\| \leq \|\theta_{:k}^\star - S_t\| + \|E_t\| \leq 2 \|\theta_{:k}^\star - S_t\|$, and the assumption that $\delta \lesssim \frac{1}{N} \frac{1}{\kappa}^{\frac{2N-2}{N}}$. On the other hand, we have

$$
\begin{aligned}
\|S_{t+1} - S_t\| &\leq \left\| \sum_{i=1}^N \eta \left( \frac{\theta_{:k}^\star - S_t}{\left\| \theta^\star - \prod w_j^{(t)} \right\|} + \delta_{i,:k} \right) \left( \prod_{j\neq i} w_{j,:k}^{(t)} \right)^2 \right\| \\
&\leq 2N\eta\sqrt{k}(\theta_1^\star)^{\frac{2(N-1)}{N}}.
\end{aligned}
\tag{83}
$$

Hence, we conclude that within $\mathcal{O}\left( \frac{\sqrt{k}\delta}{N\eta} \frac{1}{\kappa}^{\frac{2(N-1)}{N}} \right)$ iterations, we have $\|\theta_{:k}^\star - S_t\| \lesssim \sqrt{d^2 m}\alpha \vee N\eta(\theta_1^\star)^{\frac{2(N-1)}{N}}$. Overall, the total iteration complexity is bounded by $\mathcal{O}\left( \frac{k^{\frac{3}{2}}}{N\eta} \alpha^{-\frac{N-2}{N}} \right)$.

**Residual Dynamics**

Similar to the 2-layer model, here we study the surrogate of the residual term $\sum_{i=1}^N \left( w_{i,l}^{(t)} \right)^2$ for $l \geq k+1$. To this goal, we first notice that

$$
\begin{aligned}
\sum_{i=1}^N \left( w_{i,l}^{(t+1)} \right)^2 &= \sum_{i=1}^N \left( w_{i,l}^{(t)} \right)^2 + 2N\eta \left( \frac{-\prod w_{j,l}^{(t)}}{\left\| \theta^\star - \prod w_j^{(t)} \right\|} + \delta_{i,l} \right) \prod w_{j,l}^{(t)} \\
&\quad + \eta^2 \sum_{i=1}^N \left( \frac{-\prod w_{j,l}^{(t)}}{\left\| \theta^\star - \prod w_j^{(t)} \right\|} + \delta_{i,l} \right)^2 \left( \prod_{j\neq i} w_{j,l}^{(t)} \right)^2 \\
&\leq \sum_{i=1}^N \left( w_{i,l}^{(t)} \right)^2 + 4N\eta\delta \prod w_{j,l}^{(t)} \\
&\leq \sum_{i=1}^N \left( w_{i,l}^{(t)} \right)^2 + 4N\eta\delta \left( \frac{\sum_{i=1}^N \left( w_{i,l}^{(t)} \right)^2}{N} \right)^{\frac{N}{2}}.
\end{aligned}
\tag{84}
$$

Hence, once we set $z_t = \sum_{i=1}^{N} \left( w_{i,l}^{(t)} \right)^2$, we have the following simplified dynamic

$$z_{t+1} \leq z_t + 4N\eta\delta \left( \frac{z_t}{N} \right)^{\frac{N}{2}}, \tag{85}$$

with $z_0 = \Theta \left( N\alpha^{\frac{2}{N}} \right)$. We claim that within $\mathcal{O} \left( \frac{1}{N\eta\alpha} \right)$ iterations, we still have $z_t = \Theta \left( N\alpha^{\frac{2}{N}} \right)$. To show this, we suppose without loss of generality that $z_0 = N\alpha^{\frac{2}{N}}$, and define $T$ as the first time that $z_T \geq 2N\alpha^{\frac{2}{N}}$. For any $0 \leq t \leq T - 1$, we have

$$z_{t+1} \leq z_t + 4N\eta\delta 2^{\frac{N}{2}}\alpha. \tag{86}$$

We conclude that

$$T \geq \frac{N\alpha^{\frac{2}{N}}}{4N\eta\delta 2^{\frac{N}{2}}\alpha} = \frac{1}{4\delta 2^{\frac{N}{2}}} \frac{1}{\eta} \alpha^{-\frac{N-2}{N}} \gtrsim \frac{1}{N\eta} \alpha^{-\frac{N-2}{N}}. \tag{87}$$

Therefore, via a basic inequality, we have

$$\prod w_{i,l}^{(t)} \leq \left( \frac{\sum_{i=1}^{N} \left( w_{i,l}^{(t)} \right)^2}{N} \right)^{\frac{N}{2}} \lesssim \alpha. \tag{88}$$

Combining the analysis of both signal and residual terms, we conclude that within $\Theta \left( \frac{1}{N\eta} \alpha^{-\frac{N-2}{N}} \right)$ iterations, we have

$$\left\| \prod w_i^{(t)} - \theta^\star \right\| \lesssim \sqrt{d^2 m}\alpha \vee (\eta\theta_1^\star)^{\frac{2(N-1)}{N}}. \tag{89}$$

**Long Time Guarantee**

Similar to the proof of the 2-layer model, one can show that the residual term becomes the dominant term in the generalization error, and it stays in the order of $\alpha$ within $\Omega \left( \frac{1}{N\eta\delta} \alpha^{-\frac{N-2}{N}} \right)$ iterations. The details are omitted for brevity.

**Balanced Property**

To prove the balanced property, we first study the dynamic of $w_{i,l}^{(t)} - w_{j,l}^{(t)}, \forall l \in [k], i, j \in [N]$. To this goal, we have

$$w_{i,l}^{(t+1)} - w_{j,l}^{(t+1)} = \left( w_{i,l}^{(t)} - w_{j,l}^{(t)} \right) \left( 1 - \eta \left( \frac{\theta_l^\star - \prod w_{j,l}^{(t)}}{\left\| \theta^\star - \prod w_j^{(t)} \right\|} + \delta_l \right) \prod_{f \neq i,j} w_{f,l}^{(t)} \right), \tag{90}$$

which in turn implies

$$\left| w_{i,l}^{(t+1)} - w_{j,l}^{(t+1)} \right| \leq \left| w_{i,l}^{(t)} - w_{j,l}^{(t)} \right| \left( 1 - \eta \left( \frac{\theta^\star - \prod w_j^{(t)}}{\left\| \theta^\star - \prod w_j^{(t)} \right\|} + \delta_l \right) \prod_{f \neq i,j} w_f^{(t)} \right). \tag{91}$$

If $\prod w_{j,l}^{(t)} \leq \theta_l^\star - \delta \left\| \theta^\star \right\|$, the above inequality can be simplified as

$$\left| w_{i,l}^{(t)} - w_{j,l}^{(t)} \right| \leq \left| w_{i,l}^{(0)} - w_{j,l}^{(0)} \right| \lesssim \alpha^{\frac{1}{N}}, \forall i, j \in [N], l \in [k]. \tag{92}$$

Once $\prod w_{j,l}^{(t)} \geq \theta_l^\star - \delta \left\| \theta^\star \right\|$, we immediately have $w_{j,l}^{(t)} = \sqrt[N]{\theta_l^\star} \pm \mathcal{O}(\sqrt[N]{\alpha})$. Then, we show that $w_{j,l}^{(t)}$ will stay close to $\sqrt[N]{\theta_l^\star}$. To this goal, we first observe that $\left( w_{i,l}^{(t+1)} - w_{i,l}^{(t)} \right) w_{i,l}^{(t)} \equiv \left( w_{j,l}^{(t+1)} - w_{j,l}^{(t)} \right) w_{j,l}^{(t)}, \forall i, j \in [N]$, which indicates that $w_{i,l}^{(t)}$ increases or decreases simultaneously. Hence, we conclude that $\left| w_{i,l}^{(t)} - w_{j,l}^{(t)} \right| \lesssim \delta \sqrt[N]{\theta_l^\star}$.

For the residual term, we can derive a tighter bound. First, we have

$$
\left(w_{i,l}^{(t+1)}\right)^2 = \left(w_{i,l}^{(t)}\right)^2 + 2\eta\left(\frac{-\prod w_{j,l}^{(t)}}{\left\|\theta^\star - \prod w_j^{(t)}\right\|} + \delta_l\right)\prod w_{j,l}^{(t)} + \eta^2\left(\frac{-\prod w_{j,l}^{(t)}}{\left\|\theta^\star - \prod w_j^{(t)}\right\|} + \delta_l\right)^2\left(\prod_{j\neq i} w_{j,l}^{(t)}\right)^2.
\tag{93}
$$

Since we have already shown that $\prod w_{i,l}^{(t)} \lesssim \alpha$, we further have

$$
\left(w_{i,l}^{(t+1)}\right)^2 \le \left(w_{i,l}^{(t)}\right)^2 + 4\eta\delta\alpha.
\tag{94}
$$

Therefore, one can write $\left(w_{i,l}^{(t)}\right)^2 \lesssim \left(w_{i,l}^{(0)}\right)^2 + 4\eta\delta\alpha\frac{1}{\eta}\alpha^{-\frac{N-2}{N}} \lesssim \alpha^{\frac{2}{N}}$, which in turn implies $\left|w_{i,l}^{(t)} - w_{j,l}^{(t)}\right| \le \left|w_{i,l}^{(t)}\right| + \left|w_{j,l}^{(t)}\right| \lesssim \alpha^{1/N}$.

**Convergence in Under-parameterized Regime**

Similar to the 2-layer model, we consider the dynamic of $E_t = \prod w_{i,k+1:}^{(t)}$, which is characterized as follows

$$
\begin{aligned}
\|E_{t+1}\| &\le \left\| E_t + \sum_{i=1}^{N}\eta\left(\frac{-E_t}{\left\|\theta^\star - \prod w_j^{(t)}\right\|} + \delta_{k+1:}\right)\left(\prod_{j\neq i} w_{j,k+1:}^{(t)}\right)^2 \right\| \\
&\le \left\| E_t + N\eta\left(\frac{-E_t}{\left\|\theta^\star - \prod w_j^{(t)}\right\|} + \delta\right)E_t^{\frac{2(N-1)}{N}} \right\| \\
&\le \left\| E_t + N\eta\left(\frac{-E_t}{\|E_t\|} + \delta\right)E_t^{\frac{2(N-1)}{N}} \right\| \\
&\le \|E_t\| - N\eta d^{-\frac{N-1}{N}}\|E_t\|^{\frac{2N-2}{N}} + N\eta\delta\left\|E_t^{\frac{2(N-1)}{N}}\right\| \\
&\le \|E_t\| - N\eta d^{-\frac{N-1}{N}}\|E_t\|^{\frac{2N-2}{N}} + N\eta\delta\|E_t\|^{\frac{2N-2}{N}} \\
&\le \|E_t\| - N\eta d^{-\frac{N-1}{N}}\|E_t\|^{\frac{2N-2}{N}}.
\end{aligned}
\tag{95}
$$

The last inequality comes from the fact that $\delta \lesssim d^{-\frac{N-1}{N}}$ since we assume $m \gtrsim \frac{d^{\frac{2N-2}{N}}}{(1-p)^2}$. Hence, we have

$$
\|E_t\| \lesssim \left(\frac{1}{N\eta d^{-(N-1)/N}(t-\bar{T}) + 1/\|E_{\bar{T}}\|}\right)^{\frac{N}{N-2}}.
\tag{96}
$$

Since the residual term is the dominant term in the generalization error, we have

$$
\left\|\prod w_i^{(t)} - \theta^\star\right\| \lesssim \left(\frac{\left\|\prod w_i^{(\bar{T})} - \theta^\star\right\|}{\left\|\prod w_i^{(\bar{T})} - \theta^\star\right\| N\eta d^{-(N-1)/N}(t-\bar{T}) + 1}\right)^{\frac{N}{N-2}},
\tag{97}
$$

which completes the proof.

# E   Proof of Proposition 1

First, we provide an upper bound of the covering number for the $(k, \vartheta)$-approximate sparse unit ball. We defer a preliminary discussion on covering number to Appendix H.

**Lemma 4.** *Let $\mathcal{T}_{k,\vartheta} := \{u \in \mathbb{R}^d : u \text{ is } (k, \vartheta)\text{-approximate sparse}, \|u\| \le 1\}$. Then its covering number $N(\mathcal{T}_{k,\vartheta}, \varepsilon, \|\cdot\|)$ is upper bounded by*

$$
N(\mathcal{T}_{k,\vartheta}, \varepsilon, \|\cdot\|) \le \left(\frac{ed}{k}\right)^k\left(1 + \frac{4}{\varepsilon}\right)^k,
\tag{98}
$$

*provided that $\varepsilon \ge \vartheta$.*

The next lemma will play a crucial role in proving Proposition 1.

**Lemma 5.** *Suppose $x \in \mathbb{R}^d$ is a standard Gaussian vector, i.e., $x_i \overset{i.i.d.}{\sim} \mathcal{N}(0,1)$, and the noise $\varepsilon$ satisfies Assumption 1, then we have*

$$\varphi(u) = \frac{\mathbb{E}\left[\mathrm{Sign}\left(\langle x, u\rangle + \varepsilon\right)\langle x, v\rangle\right]}{\left\langle \frac{u}{\|u\|}, v\right\rangle} = \sqrt{\frac{2}{\pi}}(1-p) + \sqrt{\frac{2}{\pi}}p\mathbb{E}\left[e^{-\varepsilon^2/(2\|u\|^2)}\right].$$

The proof of this lemma can be found in Appendix G.1. Now, we are ready to prove Proposition 1. Our goal is to show that for arbitrary $u \in \mathcal{A}$, the following inequality holds

$$\left\|\frac{1}{m}\sum_{i=1}^{m}\mathrm{Sign}\left(\langle x_i, u\rangle + \varepsilon_i\right)x_i - \varphi(u)\frac{u}{\|u\|}\right\|_\infty \leq \delta \tag{99}$$

with probability at least $1 - Ce^{-cm\delta^2}$ provided that $m \gtrsim \frac{k\log(d)\log(R)\log(\frac{1}{\vartheta})}{(1-p)^2}$. Here we define $\mathcal{A} := \{u : r \leq \|u\| \leq R, u \text{ is } (k,\vartheta)\text{-approximate sparse}\}$, where $r \gtrsim \sqrt{dm/k}\vartheta\log(1/\vartheta)$. Moreover, we define $\mathcal{B} := \{u : r \leq \|u\| \leq R, \|u\|_0 \leq k\}$ and $\mathcal{C} := \{(u, u') : u \in \mathcal{A}, v \in \mathcal{B}_\zeta, \|u - u'\| \leq \zeta\}$. Here $\mathcal{B}_\zeta$ is the $\zeta$-net of $\mathcal{B}$ with $\zeta \gtrsim r$. Finally, we define $\mathcal{D} := \{\pm \mathbf{e}_j\}_{j \in [d]}$, where $\mathbf{e}_j$ forms the standard basis of $\mathbb{R}^d$. Based on these definitions, we have

$$\sup_{u \in \mathcal{A}}\left\|\frac{1}{m}\sum_{i=1}^{m}\mathrm{Sign}\left(\langle x_i, u\rangle + \varepsilon_i\right)x_i - \varphi(u)\frac{u}{\|u\|}\right\|_\infty$$

$$= \sup_{u \in \mathcal{A}, v \in \mathcal{D}}\frac{1}{m}\sum_{i=1}^{m}\mathrm{Sign}\left(\langle x_i, u\rangle + \varepsilon_i\right)\langle x_i, v\rangle - \frac{\varphi(u)}{\|u\|}\langle u, v\rangle \tag{100}$$

$$= \sup_{v \in \mathcal{D}}\left\{\sup_{u \in \mathcal{A}}\frac{1}{m}\sum_{i=1}^{m}\mathrm{Sign}\left(\langle x_i, u\rangle + \varepsilon_i\right)\langle x_i, v\rangle - \frac{\varphi(u)}{\|u\|}\langle u, v\rangle\right\}.$$

We then show that for each element $y \in \mathcal{D}$, $\sup_{u \in \mathcal{A}}\frac{1}{m}\sum_{i=1}^{m}\mathrm{Sign}\left(\langle x_i, u\rangle + \varepsilon_i\right)\langle x_i, y\rangle - \varphi(u)\frac{\langle u, y\rangle}{\|u\|}$, $j \in [d]$ is $\mathcal{O}\left(\frac{1}{m}\right)$-sub-Gaussian random variable. To see this, note that

$$\left\|\mathrm{Sign}\left(\langle x_i, u\rangle + \varepsilon_i\right)x_{i,j} - \varphi(u)\frac{u_j}{\|u\|}\right\|_{\psi_2} \leq \left\|\mathrm{Sign}\left(\langle x_i, u\rangle + \varepsilon_i\right)x_{i,j}\right\|_{\psi_2} + \sqrt{\frac{2}{\pi}}$$

$$\leq \|x_{i,j}\|_{\psi_2} + \sqrt{\frac{2}{\pi}} = \mathcal{O}(1). \tag{101}$$

Here we use the property of sub-Gaussian norm. This implies that $\frac{1}{m}\sum_{i=1}^{m}\mathrm{Sign}\left(\langle x_i, u\rangle + \varepsilon_i\right)x_{i,j} - \varphi(u)\frac{u_j}{\|u\|}$ is $\mathcal{O}\left(\frac{1}{m}\right)$-sub-Gaussian random variable, since it is the sample average of $\mathrm{Sign}\left(\langle x_i, u\rangle + \varepsilon_i\right)x_{i,j} - \varphi(u)\frac{u_j}{\|u\|}$.

Hence, via maximal inequality, we have that for $\forall t > 0$,

$$\mathbb{P}\left(\sup_{u \in \mathcal{A}}\left\|\frac{1}{m}\sum_{i=1}^{m}\mathrm{Sign}\left(\langle x_i, u\rangle + \varepsilon_i\right)x_i - \varphi(u)\frac{u}{\|u\|}\right\|_\infty\right.$$

$$\left.\geq \sup_{y \in \mathcal{D}}\mathbb{E}\left[\sup_{u \in \mathcal{A}}\frac{1}{m}\sum_{i=1}^{m}\mathrm{Sign}\left(\langle x_i, u\rangle + \varepsilon_i\right)\langle x_i, y\rangle - \varphi(u)\frac{\langle u, y\rangle}{\|u\|}\right] + t\right) \tag{102}$$

$$\leq 2de^{-cmt^2}.$$

Hence, it suffices to study $\mathbb{E}\left[\sup_{u \in \mathcal{A}}\frac{1}{m}\sum_{i=1}^{m}\mathrm{Sign}\left(\langle x_i, u\rangle + \varepsilon_i\right)x_{i,1} - \varphi(u)\frac{u_1}{\|u\|}\right]$. To this goal, we decompose it into two terms via triangle inequality.

$$\mathbb{E}\left[\sup_{u \in \mathcal{A}}\frac{1}{m}\sum_{i=1}^{m}\mathrm{Sign}\left(\langle x_i, u\rangle + \varepsilon_i\right)x_{i,1} - \varphi(u)\frac{u_1}{\|u\|}\right] \leq (A) + (B), \tag{103}$$

where

$$(A) := \mathbb{E}\left[\sup_{u \in \mathcal{B}_\zeta} \frac{1}{m}\sum_{i=1}^{m} \mathrm{Sign}\left(\langle x_i, u\rangle + \varepsilon_i\right) x_{i,1} - \varphi(u)\frac{u_1}{\|u\|}\right], \tag{104}$$

and

$$(B) := \mathbb{E}\left[\sup_{(u,u') \in \mathcal{C}} \frac{1}{m}\sum_{i=1}^{m} \left(\mathrm{Sign}\left(\langle x_i, u\rangle + \varepsilon_i\right) - \mathrm{Sign}\left(\langle x_i, u'\rangle + \varepsilon_i\right)\right) x_{i,1} - \varphi(u)\frac{u_1}{\|u\|} + \varphi(u')\frac{u'_1}{\|u'\|}\right]. \tag{105}$$

We first control (A). To this goal, we apply the union bound. Note that $\frac{1}{m}\sum_{i=1}^{m} \mathrm{Sign}\left(\langle x_i, u\rangle + \varepsilon_i\right) x_{i,1} - \varphi(u)\frac{u_1}{\|u\|}$ is $\mathcal{O}(\frac{1}{m})$-sub-Gaussian and $|\mathcal{B}_\zeta| \leq \left(\frac{R}{\zeta}\right)^{Ck\log(d)}$. We then have

$$(A) \lesssim \sqrt{\frac{k\log(d)\log\left(\frac{R}{\zeta}\right)}{m}}. \tag{106}$$

Now we control (B). Via triangle inequality, we first obtain

$$(B) \leq \underbrace{\mathbb{E}\left[\sup_{(u,u') \in \mathcal{C}} \frac{1}{m}\sum_{i=1}^{m} \left(\mathrm{Sign}\left(\langle x_i, u\rangle + \varepsilon_i\right) - \mathrm{Sign}\left(\langle x_i, u'\rangle + \varepsilon_i\right)\right) x_{i,1}\right]}_{(B_1)} \tag{107}$$
$$+ \underbrace{\sup_{(u,u') \in \mathcal{C}} \left\{-\varphi(u)\frac{u_1}{\|u\|} + \varphi(u')\frac{u'_1}{\|u'\|}\right\}}_{(B_2)}.$$

For the first part, applying Hölder's inequality leads to

$$(B_1) \leq \mathbb{E}\left[\sup_{(u,u') \in \mathcal{C}} \left(\frac{1}{m}\sum_{i=1}^{m} |\mathrm{Sign}\left(\langle x_i, u\rangle + \varepsilon_i\right) - \mathrm{Sign}\left(\langle x_i, u'\rangle + \varepsilon_i\right)|\right) \max_{1 \leq i \leq m} |x_{i,1}|\right]$$
$$\leq \mathbb{E}\left[\sup_{(u,u') \in \mathcal{C}} \left(\frac{1}{m}\sum_{i=1}^{m} \mathbb{1}\left(|\langle x_i, u - u'\rangle| \geq |\langle x_i, u\rangle + \varepsilon_i|\right)\right) \max_{1 \leq i \leq m} |x_{i,1}|\right]$$
$$\leq \underbrace{\mathbb{E}\left[\sup_{\|\Delta u\| \leq \zeta} \left(\frac{1}{m}\sum_{i=1}^{m} \mathbb{1}\left(|\langle x_i, \Delta u\rangle| \geq t\right)\right) \max_{1 \leq i \leq m} |x_{i,1}|\right]}_{(B_3)} \tag{108}$$
$$+ \underbrace{\mathbb{E}\left[\sup_{u \in \mathcal{B}_\zeta} \left(\frac{1}{m}\sum_{i=1}^{m} \mathbb{1}\left(|\langle x_i, u\rangle + \varepsilon_i| \leq t\right)\right) \max_{1 \leq i \leq m} |x_{i,1}|\right]}_{(B_4)},$$

where $t > 0$ is a constant to be determined later. Here, we used the fact that $\mathbb{1}\left(|\langle x_i, u - u'\rangle| \geq |\langle x_i, u\rangle + \varepsilon_i|\right) \leq \mathbb{1}\left(|\langle x_i, \Delta u\rangle| \geq t\right) + \mathbb{1}\left(|\langle x_i, u\rangle + \varepsilon_i| \leq t\right)$ in the last inequality. We first bound $(B_3)$

$$(B_3) \leq \mathbb{E}\left[\left(\frac{1}{m}\sum_{i=1}^{m} \mathbb{1}\left(\zeta\|x_i\| \geq t\right)\right) \max_{1 \leq i \leq m} |x_{i,1}|\right]$$
$$\leq \mathbb{E}\left[\mathbb{1}\left(\zeta\|x_i\| \geq t\right)\right] \mathbb{E}\left[\max_{j \neq i} |x_{j,1}|\right] + \mathbb{E}\left[\mathbb{1}\left(\zeta\|x_i\| \geq t\right) |x_{i,1}|\right] \tag{109}$$
$$\lesssim e^{-C\frac{t^2}{\zeta^2}}\sqrt{\log(m)} + \mathbb{E}\left[\mathbb{1}\left(\zeta\|x_i\| \geq t\right) |x_{i,1}|\right],$$

provided that $\frac{t}{\zeta} \gtrsim \sqrt{d}$. Applying Cauchy-Schwarz inequality, we have

$$\mathbb{E}\left[\mathbb{1}\left(\zeta\|x_i\| \geq t\right) |x_{i,1}|\right] \leq \sqrt{\mathbb{P}\left(\zeta\|x_i\| \geq t\right)}\sqrt{\mathbb{E}\left[x_{i,1}^2\right]} \leq e^{-C\frac{t^2}{\zeta^2}}. \tag{110}$$

Hence, we conclude that $(B_3) \lesssim e^{-C\frac{t^2}{\zeta^2}}\sqrt{\log(m)}$. Next we control $(B_4)$. Note that $\max_i |x_{i,1}|$ is $\mathcal{O}(\log(m))$-sub-Gaussian. Via union bound, we have

$$(B_4) \leq \sup_{u \in \mathcal{B}} \mathbb{E}\left[\mathbb{1}\left(|\langle x_i, u\rangle + \varepsilon_i| \leq t\right) \max_{1 \leq i \leq m} |x_{i,1}|\right] + C\sqrt{\frac{k\log(m)\log(d)\log(\frac{R}{\zeta})}{m}}. \tag{111}$$

For the first part, applying the similar decomposition method, we have

$$\mathbb{E}\left[\mathbb{1}\left(|\langle x_i, u\rangle + \varepsilon_i| \leq t\right) \max_{1 \leq i \leq m} |x_{i,1}|\right] \leq \mathbb{E}\left[\mathbb{1}\left(|\langle x_i, u\rangle + \varepsilon_i| \leq t\right) \max_{j \neq i} |x_{i,1}|\right]$$
$$+ \mathbb{E}\left[\mathbb{1}\left(|\langle x_i, u\rangle + \varepsilon_i| \leq t\right) |x_{i,1}|\right] \tag{112}$$
$$\lesssim \sqrt{\log(m)}\frac{t}{r}.$$

Hence, we conclude that $(B_4) \lesssim \sqrt{\log(m)}\frac{t}{r} + \sqrt{\frac{k\log(m)\log(d)\log(\frac{R}{\zeta})}{m}}$. For $(B_2)$, we first have

$$\left|-\varphi(u)\frac{u_1}{\|u\|} + \varphi(u')\frac{u_1'}{\|u'\|}\right| = |\varphi(u') - \varphi(u)|\frac{|u_1|}{\|u\|} + \varphi(u')\left|\frac{u_1'}{\|u'\|} - \frac{u_1}{\|u\|}\right| \tag{113}$$
$$\lesssim |\varphi(u') - \varphi(u)| + \zeta.$$

For the first part, we use Mean Value Theorem to write

$$|\varphi(u') - \varphi(u)| \leq \|\nabla\varphi(v)\| \|u' - u\| \leq \|\nabla\varphi(v)\|\zeta, \tag{114}$$

where $v$ is a point between $u$ and $u'$. Note that $\nabla\varphi(v) = \sqrt{\frac{2}{\pi}}p\mathbb{E}\left[\frac{\varepsilon^2 v}{\|v\|^4}e^{-\frac{\varepsilon^2}{2\|v\|^2}}\right]$. Hence, we have

$$\sup_{\|v\| \geq r} \|\nabla\varphi(v)\| \lesssim \sup_{\|v\| \geq r} \mathbb{E}\left[\frac{\varepsilon^2}{\|v\|^3}e^{-\frac{\varepsilon^2}{2\|v\|^2}}\right] \leq \frac{1}{r}\sup_{\|v\| \geq r}\mathbb{E}\left[\frac{\varepsilon^2}{\|v\|^2}e^{-\frac{\varepsilon^2}{2\|v\|^2}}\right] \lesssim \frac{1}{r}. \tag{115}$$

Overall, we have $(B_2) \lesssim \frac{\zeta}{r}$, which results in

$$\mathbb{E}\left[\sup_{u \in \mathcal{A}} \frac{1}{m}\sum_{i=1}^m \text{Sign}\left(\langle x_i, u\rangle + \varepsilon_i\right)x_{i,1} - \varphi(u)\frac{u_1}{\|u\|}\right]$$
$$\lesssim \frac{\zeta}{r} + e^{-C\frac{t^2}{\zeta^2}}\sqrt{\log(m)} + \sqrt{\log(m)}\frac{t}{r} + \sqrt{\frac{k\log(m)\log(d)\log(\frac{R}{\zeta})}{m}}. \tag{116}$$

Hence, once we set $\zeta \asymp \vartheta$, and $t \asymp \sqrt{d}\vartheta\log(m)$, together with the assumption that $r \gtrsim \sqrt{\frac{dm}{k}}\vartheta\log\left(\frac{1}{\vartheta}\right)$, we conclude that

$$\mathbb{E}\left[\sup_{u \in \mathcal{A}} \frac{1}{m}\sum_{i=1}^m \text{Sign}\left(\langle x_i, u\rangle + \varepsilon_i\right)x_{i,1} - \varphi(u)\frac{u_1}{\|u\|}\right] \lesssim \sqrt{\frac{k\log^2(m)\log(d)\log(\frac{R}{\vartheta})}{m}}. \tag{117}$$

This leads to

$$\mathbb{P}\left(\sup_{u \in \mathcal{A}} \left\|\frac{1}{m}\sum_{i=1}^m \text{Sign}\left(\langle x_i, u\rangle + \varepsilon_i\right)x_i - \varphi(u)\frac{u}{\|u\|}\right\|_\infty \geq C\sqrt{\frac{k\log^2(m)\log(d)\log(\frac{R}{\vartheta})}{m}} + \delta\right)$$
$$\leq 2de^{-cm\delta^2}. \tag{118}$$

Therefore, the following inequality holds, provided that $m \gtrsim \frac{k\log^2(m)\log(d)\log(\frac{R}{\vartheta})}{(1-p)^2\delta^2}$

$$\mathbb{P}\left(\sup_{u \in \mathcal{A}} \left\|\frac{1}{\varphi(u)}\frac{1}{m}\sum_{i=1}^m \text{Sign}\left(\langle x_i, u\rangle + \varepsilon_i\right)x_i - \frac{u}{\|u\|}\right\|_\infty \geq \delta\right) \leq e^{-cm\delta^2}. \tag{119}$$

Now we turn to the case $m \gtrsim \frac{d}{(1-p)^2}$. Following the same technique, it suffices to bound $\mathbb{E}\left[\sup_{u \in \mathbb{R}^d} \frac{1}{m} \sum_{i=1}^m \mathrm{Sign}\left(\langle x_i, u\rangle + \varepsilon_i\right) x_{i,1} - \varphi(u)\frac{u_1}{\|u\|}\right]$. To this goal, we first notice that

$$
\begin{aligned}
&\mathbb{E}\left[\sup_{u \in \mathbb{R}^d} \frac{1}{m} \sum_{i=1}^m \mathrm{Sign}\left(\langle x_i, u\rangle + \varepsilon_i\right) x_{i,1} - \varphi(u)\frac{u_1}{\|u\|}\right] \\
&= \mathbb{E}\left[\underbrace{\sup_{\|u\|=1, \lambda \in \mathbb{R}} \frac{1}{m} \sum_{i=1}^m \mathrm{Sign}\left(\langle x_i, u\rangle + \lambda\varepsilon_i\right) x_{i,1} - \varphi(u)u_1}_{(A)}\right].
\end{aligned}
\tag{120}
$$

Similarly, applying one-step discretization, we have

$$
\begin{aligned}
(A) \leq &\mathbb{E}\left[\underbrace{\sup_{u \in \mathbb{S}_\varepsilon, \lambda \in \mathbb{R}} \frac{1}{m} \sum_{i=1}^m \mathrm{Sign}\left(\langle x_i, u\rangle + \lambda\varepsilon_i\right) x_{i,1} - \phi(\lambda)u_1}_{(B)}\right] \\
&+ \mathbb{E}\left[\underbrace{\sup_{\|u-u'\|\leq\varepsilon, \lambda \in \mathbb{R}} \frac{1}{m} \sum_{i=1}^m \left(\mathrm{Sign}\left(\langle x_i, u\rangle + \lambda\varepsilon_i\right) - \mathrm{Sign}\left(\langle x_i, u'\rangle + \lambda\varepsilon_i\right)\right) x_{i,1} + \phi(\lambda)\left(u_1' - u_1\right)}_{(C)}\right].
\end{aligned}
\tag{121}
$$

Here $\phi(\lambda) = \sqrt{\frac{2}{\pi}}(1-p) + \sqrt{\frac{2}{\pi}}p\mathbb{E}\left[e^{-\lambda^2\varepsilon^2/2}\right]$ is the same as before. We first control $(B)$. To this goal, we show that $\sup_{\lambda \in \mathbb{R}} \frac{1}{m} \sum_{i=1}^m \mathrm{Sign}\left(\langle x_i, u\rangle + \lambda\varepsilon_i\right) x_{i,1} - \phi(\lambda)u_1$ is $\mathcal{O}(1/m)$-sub-Gaussian. We prove it via checking the sub-Gaussian norm

$$
\left\|\sup_{\lambda \in \mathbb{R}} \mathrm{Sign}\left(\langle x_i, u\rangle + \lambda\varepsilon_i\right) x_{i,1} - \phi(\lambda)u_1\right\|_{\psi_2} \leq \|\|x_{i,1}\|\|_{\psi_2} + \sqrt{\frac{2}{\pi}} = \mathcal{O}(1).
\tag{122}
$$

Hence, via maximum inequality, we have

$$
(B) \leq \mathbb{E}\left[\underbrace{\sup_{\lambda \in \mathbb{R}} \frac{1}{m} \sum_{i=1}^m \mathrm{Sign}\left(\langle x_i, u\rangle + \lambda\varepsilon_i\right) x_{i,1} - \phi(\lambda)u_1}_{(D)}\right] + \mathcal{O}\left(\sqrt{\frac{d\log\left(\frac{1}{\varepsilon}\right)}{m}}\right).
\tag{123}
$$

To control $(D)$, we further decompose it into two parts,

$$
\begin{aligned}
(D) \leq &\mathbb{E}\left[\underbrace{\sup_{\nu \in [0,1]} \frac{1}{m} \sum_{i=1}^m \mathrm{Sign}\left(\nu\langle x_i, u\rangle + \varepsilon_i\right) x_{i,1} - \phi\left(\frac{1}{\nu}\right)u_1}_{(D_1)}\right] \\
&+ \mathbb{E}\left[\underbrace{\sup_{\lambda \in [0,1]} \frac{1}{m} \sum_{i=1}^m \mathrm{Sign}\left(\langle x_i, u\rangle + \lambda\varepsilon_i\right) x_{i,1} - \phi(\lambda)u_1}_{(D_2)}\right].
\end{aligned}
\tag{124}
$$

To control $(D_1)$ and $(D_2)$ we use arguments based on bracketing maximal inequality. We defer a preliminary discussion on bracketing maximal inequality to Appendix H. We first control $(D_1)$. Let $\mathbb{T}_\xi$ be defined as the $\xi$-net of the interval $[0,1]$. We show that for any $\nu, \nu' \in [0,1]$ such that $|\nu - \nu'| \leq \xi$, we can control $\|(\mathrm{Sign}(\nu\langle x_i, u\rangle + \varepsilon_i) - \mathrm{Sign}(\nu'\langle x_i, u\rangle + \varepsilon_i))x_{i,1}\|_{L_2(\mathbb{P})}$. To this goal, we first have

$$
\begin{aligned}
&\mathbb{E}\left[\left(\mathrm{Sign}(\nu\langle x_i, u\rangle + \varepsilon_i) - \mathrm{Sign}(\nu'\langle x_i, u\rangle + \varepsilon_i)\right)^2 x_{i,1}^2\right] \\
&\lesssim \mathbb{E}\left[|\mathrm{Sign}(\nu\langle x_i, u\rangle + \varepsilon_i) - \mathrm{Sign}(\nu'\langle x_i, u\rangle + \varepsilon_i)|\right] \\
&\leq \mathbb{E}\left[\mathbb{1}\left(|(\nu - \nu')\langle x_i, u\rangle| \geq t\right) + \mathbb{1}\left(|\nu\langle x_i, u\rangle + \varepsilon_i| \leq t\right)\right] \\
&\lesssim e^{-C\frac{t^2}{\xi^2}} + t.
\end{aligned}
\tag{125}
$$

Upon picking $t \asymp \xi \log\left(\frac{1}{\xi}\right)$, we have

$$\|(\text{Sign}(\nu \langle x_i, u \rangle + \varepsilon_i) - \text{Sign}(\nu' \langle x_i, u \rangle + \varepsilon_i)) x_{i,1}\|_{L_2(\mathbb{P})} \lesssim \sqrt{\xi \log\left(\frac{1}{\xi}\right)}. \tag{126}$$

Therefore, the bracketing number is bounded by $N_{[]}(\varepsilon \|F\|, \mathcal{F}, \|\cdot\|) \lesssim C\frac{1}{\sqrt{\varepsilon}}$, which in turn leads to an upper bound on the bracketing entropy $J_{[]}(1, \mathcal{F}, L_2(\mathbb{P})) \lesssim 1$. Applying Theorem 6 leads to

$$(D_1) \lesssim \sqrt{\frac{1}{m}}. \tag{127}$$

Similarly, we can show that $(D_2) \lesssim \sqrt{\frac{1}{m}}$. Therefore, we conclude that

$$(B) \lesssim \sqrt{\frac{d \log\left(\frac{1}{\varepsilon}\right)}{m}}. \tag{128}$$

For $(C)$, we can use the similar technique in the overparameterized setting ($m \ll d$), which leads to

$$(C) \lesssim \sqrt{\log(m)}\varepsilon + \sqrt{\frac{d \log(m)}{m}}. \tag{129}$$

Therefore, once we set $\varepsilon \asymp \sqrt{\frac{d}{m}}$, we immediately obtain

$$(A) \lesssim \sqrt{\frac{d \log(m)}{m}}. \tag{130}$$

Combining the derived bounds results in

$$\mathbb{P}\left(\sup_{u \in \mathbb{R}^d} \left\| \frac{1}{m} \sum_{i=1}^m \text{Sign}\left(\langle x_i, u \rangle + \varepsilon_i\right) x_i - \varphi(u)\frac{u}{\|u\|} \right\|_\infty \geq C\sqrt{\frac{d \log(m)}{m}} + \delta \right) \leq e^{c_1 \log(d) - c_2 m \delta^2}. \tag{131}$$

Assuming $m \gtrsim \frac{d \log(m)}{(1-p)^2}$, the above bound reduces to

$$\mathbb{P}\left(\sup_{u \in \mathbb{R}^d} \left\| \frac{1}{\varphi(u)} \frac{1}{m} \sum_{i=1}^m \text{Sign}\left(\langle x_i, u \rangle + \varepsilon_i\right) x_i - \frac{u}{\|u\|} \right\|_\infty \geq \delta \right) \leq e^{-cm\delta^2}. \tag{132}$$

## F   Auxiliary Lemmas

**Lemma 6.** *Suppose $x_1, \cdots, x_m$ are i.i.d. standard Gaussian vectors with dimension $d$. Then, for arbitrary $\delta > 0$ we have*

$$\mathbb{P}\left(\sup_{\|u\|=1} \left| \frac{1}{m} \sum_{i=1}^m |\langle x_i, u \rangle| - \sqrt{\frac{2}{\pi}} \right| \geq C\sqrt{\frac{d}{m}} + \delta \right) \leq e^{-cm\delta^2}. \tag{133}$$

*Here $C, c$ are universal constants.*

*Proof.* This lemma directly follows from the standard expectation and high probability bounds for sub-Gaussian process. See e.g., [29, Lemma 4] for a simple proof. $\square$

**Lemma 7.** *For two arbitrary vectors $a, b \in \mathbb{R}^n$, we have*

$$\|a \odot b\| \leq \|a\|_\infty \|b\|. \tag{134}$$

# G  Deferred Proofs

## G.1  Proof of Lemma 5

*Proof.* To prove this lemma, it suffices to show that, for any $u, v \in \mathbb{R}^d$, we have

$$\mathbb{E}\left[\text{Sign}\left(\varepsilon + \langle x, u \rangle\right) \langle x, v \rangle\right] = \sqrt{\frac{2}{\pi}} \mathbb{E}\left[e^{-\varepsilon^2/2\|u\|^2}\right] \left\langle \frac{u}{\|u\|}, v \right\rangle. \tag{135}$$

Without loss of generality, we assume that $\|u\| = \|v\| = 1$. Let us denote $w := \langle x, u \rangle, z := \langle x, v \rangle, \rho := \text{Cov}(w, z) = \langle u, v \rangle$. Then

$$\begin{aligned}
\mathbb{E}\left[\text{Sign}\left(\varepsilon + \langle x, u \rangle\right) \langle x, v \rangle\right] &= \mathbb{E}\left[\text{Sign}\left(\varepsilon + w\right) z\right] \\
&\overset{(a)}{=} \rho \mathbb{E}\left[\text{Sign}(w + \varepsilon) w\right] \\
&= \rho \mathbb{E}_\varepsilon\left[\int_{-\varepsilon}^{\infty} t\frac{1}{\sqrt{2\pi}} e^{-t^2/2} dt - \int_{-\infty}^{-\varepsilon} t\frac{1}{\sqrt{2\pi}} e^{-t^2/2} dt\right] \\
&= \rho \mathbb{E}_\varepsilon\left[\int_{-\varepsilon}^{\infty} t\frac{1}{\sqrt{2\pi}} e^{-t^2/2} dt + \int_{\varepsilon}^{\infty} t\frac{1}{\sqrt{2\pi}} e^{-t^2/2} dt\right] \\
&= 2\rho \mathbb{E}_\varepsilon\left[\int_{|\varepsilon|}^{\infty} t\frac{1}{\sqrt{2\pi}} e^{-t^2/2} dt\right] \\
&= \sqrt{\frac{2}{\pi}} \langle u, v \rangle \mathbb{E}_\varepsilon\left[\int_{|\varepsilon|}^{\infty} d\left(-e^{-t^2/2}\right)\right] \\
&= \sqrt{\frac{2}{\pi}} \langle u, v \rangle \mathbb{E}_\varepsilon\left[e^{-\varepsilon^2/2}\right].
\end{aligned} \tag{136}$$

Here in (a) we use the fact that $z|w, \varepsilon \sim \mathcal{N}(\rho w, 1 - \rho^2)$ since $\varepsilon$ is independent of $w, z$. Hence, we have

$$\mathbb{E}\left[\text{Sign}\left(\varepsilon + \langle x, u \rangle\right) \langle x, v \rangle\right] = \sqrt{\frac{2}{\pi}} \mathbb{E}\left[e^{-\varepsilon^2/2\|u\|^2}\right] \left\langle \frac{u}{\|u\|}, v \right\rangle \tag{137}$$

for any $u, v \in \mathbb{R}^d$. On the other hand, it is easy to verify that $\mathbb{E}\left[\text{Sign}\left(\langle x, u \rangle\right) \langle x, v \rangle\right] = \sqrt{\frac{2}{\pi}} \left\langle \frac{u}{\|u\|}, v \right\rangle$. The proof is completed by noting that the corruption probability is $p$.  $\square$

# H  Preliminaries on the Uniform Concentration Bounds

In this section, we provide the preliminary probability tools for proving Proposition 1.

**Definition 2** (Sub-Gaussian random variable). *We say a random variable $X \in \mathbb{R}$ with expectation $\mathbb{E}[X] = \mu$ is $\sigma^2$-sub-Gaussian if for all $\lambda \in \mathbb{R}$, we have $\mathbb{E}\left[e^{\lambda(X-\mu)}\right] \leq e^{\frac{\lambda^2 \sigma^2}{2}}$. Moreover, the sub-Gaussian norm of $X$ is defined as $\|X\|_{\psi_2} := \sup_{p \geq 1}\left\{p^{-1/2}(\mathbb{E}[|X|^p])^{1/p}\right\}$.*

According to [39], the following statements are equivalent:

- $X$ is $\sigma^2$-sub-Gaussian.
- (Tail bound) For any $t > 0$, we have $\mathbb{P}(|X - \mu| \geq t) \leq 2e^{-\frac{t^2}{2\sigma^2}}$.
- (Moment bound) We have $\|X\|_{\psi_2} \lesssim \sigma$.

Next, we provide the definitions of the sub-Gaussian process, $\varepsilon$-net, and covering number.

**Definition 3** (Sub-Gaussian process). *A zero mean stochastic process $\{\mathcal{X}_\theta, \theta \in \mathbb{T}\}$ is a $\sigma^2$-sub-Gaussian process with respect to a metric $d$ on a set $\mathbb{T}$, if for every $\theta, \theta' \in \mathbb{T}$, the random variable $\mathcal{X}_\theta - \mathcal{X}_{\theta'}$ is $(\sigma d(\theta, \theta'))^2$-sub-Gaussian.*

**Definition 4** ($\varepsilon$-net and covering number). *A set $\mathcal{N}$ is called an $\varepsilon$-net for $(\mathbb{T}, d)$ if for every $t \in \mathbb{T}$, there exists $\pi(t) \in \mathcal{N}$ such that $d(t, \pi(t)) \leq \varepsilon$. The covering number $N(\mathbb{T}, d, \varepsilon)$ is defined as the smallest cardinality of an $\varepsilon$-net for $(\mathbb{T}, d)$:*

$$N(\mathbb{T}, d, \varepsilon) := \inf\{|\mathcal{N}| : \mathcal{N} \text{ is an } \varepsilon\text{-net for } (\mathbb{T}, d)\}.$$

**Definition 5** (Bracketing number, Definition 2.1.6 in [37]). *Given two functions $l$ and $u$, the bracket $[l, u]$ is the set of all functions $f$ with $l \leq f \leq u$. An $\varepsilon$-bracket is a bracket $[l, u]$ with $\|u - l\| < \varepsilon$. The bracketing number $N_{[]}(\varepsilon, \mathcal{F}, \|\cdot\|)$ is the minimum number of $\varepsilon$-brackets needed to cover $\mathcal{F}$. The bracketing entropy is the logarithm of the bracketing number. In the definition of the bracketing number, the upper and lower bounds $u$ and $l$ of the brackets need not belong to $\mathcal{F}$ themselves but are assumed to have finite norms.*

Bracketing number can be regarded as an analog of covering number, describing the geometric complexity of the underlining function space. Although bracketing number of a general function class is difficult to characterize, for some specific function classes, we can easily derive upper bounds for their bracketing number. In particular, we have the following result for Lipschitz functions.

**Theorem 5** (Adapted from Theorem 2.7.11 in [37]). *Let $\mathcal{F} = \{f_t : t \in T\}$ be a class of functions. Suppose that for arbitrary $s, t \in T$, we have*

$$|f_s(x) - f_t(x)| \leq d(s, t) F(x), \tag{138}$$

*for some metric $d$ on the index set, function $F$ on the sample space, and every $x$. Then, for any norm $\|\cdot\|$,*

$$N_{[]}(2\varepsilon \|F\|, \mathcal{F}, \|\cdot\|) \leq N(\varepsilon, T, d). \tag{139}$$

**Theorem 6** (Adapted from Theorem 2.14.2 in [37]). *For a given norm $\|\cdot\|$, define a bracketing integral of a class of functions $\mathcal{F}$ as*

$$J_{[]}(\delta, \mathcal{F}, \|\cdot\|) = \int_0^\delta \sqrt{1 + \log N_{[]}(\varepsilon \|F\|, \mathcal{F}, \|\cdot\|)} d\varepsilon. \tag{140}$$

*Let $\mathcal{F}$ be a class of measurable functions with measurable envelope function $F$, we have*

$$\mathbb{E}\left[\sup_{f \in \mathcal{F}} \frac{1}{n} \sum_{i=1}^n f(X_i) - \mathbb{E}[f(X)]\right] \lesssim J_{[]}(1, \mathcal{F}, L_2(\mathbb{P})) \frac{\|F\|_{L_2(\mathbb{P})}}{\sqrt{n}}, \tag{141}$$

*where $\mathbb{P}$ is the distribution of $X$, and the $L_2(\mathbb{P})$-norm is defined as $\|f\|_{L_2(\mathbb{P})} := \left(\int_\Omega f^2(\omega) d\mathbb{P}(\omega)\right)^{1/2}$.*