# OpenReview forum: "Blessing of Depth in Linear Regression: Deeper Models Have Flatter Landscape Around the True Solution"
_NeurIPS.cc/2022/Conference — NeurIPS 2022 Accept_

### Official Review · Reviewer_kMm7 · 2022-06-30

**Rating:** 7
**Confidence:** 3
**Soundness:** 3 good
**Presentation:** 3 good
**Contribution:** 4 excellent

**Summary:**

This paper studies the problem of robust and sparse linear regression under an over-parameterized setting. The authors show that this problem does not have a benign landscape. But for any N-layer model with $N \ge 2$, a simple sub-gradient with small initialization will converge to a small neighborhood of a balanced true solution, the algorithm will stay close to the ground truth for a longer time, and depth flattens the optimization landscape around the solution obtained by SubGM.

**Questions:**

What is the reference [42] in Appendix A? I think the authors should carefully the references.

Considering the similar setting but noiseless data, what would the results be?

**Limitations:**

Yes, the authors have adequately addressed the limitations and potential negative societal impact of their work

**Strengths And Weaknesses:**

Strength:

The paper studies an interesting question related to the over-parameterized problem. In particular, the authors show interesting and important findings that deeper models have provably better optimization landscape around the solution trajectory.

Paper is written well.

---

> ### Author Response · Authors · 2022-07-31
> **Response to Reviewer kMm7**
>
> We are happy that the reviewer finds our paper interesting!
>
> ### Response to questions
>
> >  "What is the reference [42] in Appendix A? I think the authors should carefully the references."
>
> We thank the reviewer for catching this mistake, which will be fixed in our revised manuscript.
>
> >  "Considering the similar setting but noiseless data, what would the results be?"
>
> We are thankful to the reviewer for this insightful question. With noiseless data, all of our results hold with corruption probability $p=0$. The only difference is that there is no longer a "bad" (unidentifiable) true solution in Theorem 1 since the right-hand side of Equation 2 is $0$. This result is analogous to the results of \[1,2\] which show that deep linear neural networks with noiseless data have no spurious local solutions.
>
> \[1\] Srinadh Bhojanapalli, Behnam Neyshabur, and Nathan Srebro. Global optimality of local search for low rank matrix recovery. *arXiv preprint arXiv:1605.07221*, 2016.
>
> \[2\] Kenji Kawaguchi. Deep learning without poor local minima. *arXiv preprint arXiv:1605.07110*, 2016.

---

> > ### Comment · Reviewer_kMm7 · 2022-08-10
> > **Sorry for the delayed response**
> >
> > I would like to thank the authors for the detailed response, which has addressed my concerns well.

---

### Official Review · Reviewer_vNfZ · 2022-07-04

**Rating:** 9
**Confidence:** 5
**Soundness:** 4 excellent
**Presentation:** 4 excellent
**Contribution:** 4 excellent

**Summary:**

This paper considers the implicit bias of the deep (diag) linear model. The previous analysis need the loss to be a l2 loss. In this paper, the author consider a robust statistic model where l1 loss should be applied. by ananlysising the landscape around the true solution, The author discovered that the true solutions are likely to be non-critical points under the one-layer convex parameterization, but have a good local property when multilayer nonconvex parameterization is applied.

**Questions:**

- The author can provide more proof initution in the main text. In the main text, the convergence result of the population dynamic and how the initialization scaling changes the stability result is missing. (I'm also curious about it)
- Can you compare the results when the network becomes deeper? It's a trade off or deeper is better?

**Strengths And Weaknesses:**

This paper is quite interesting and I strongly recommend to accept. The proof technique is also new to me. The results include converging of the sub-gradient method and long escape time after arrived the balanced solution.

---

> ### Author Response · Authors · 2022-07-31
> **Response to Reviewer vNfZ**
>
> We are glad that the reviewer finds our paper interesting!
>
> ### Response to questions
>
> >  "The author can provide more proof intuition in the main text. In the main text, the convergence result of the population dynamic and how the initialization scaling changes the stability result is missing. (I'm also curious about it)"
>
> We thank the reviewer for this suggestion. In the revised manuscript, we will further shed light on population dynamic and the role of initialization scale.
>
> >  "Can you compare the results when the network becomes deeper? It's a trade off or deeper is better?"
>
> We thank the reviewer for this insightful question. Regarding the optimization landscape, Theorem 4 shows that depth consistently flattens the optimization landscape around the balanced solution (therefore, the deeper, the better!). However, in terms of the convergence of SubGM, there exists a tradeoff: while deeper models lead to better generalization error and longer escape time, they take longer to train (see Theorems 2 and 3). We will further clarify this point in our revised manuscript.

---

### Official Review · Reviewer_ffkC · 2022-07-09

**Rating:** 5
**Confidence:** 4
**Soundness:** 1 poor
**Presentation:** 3 good
**Contribution:** 2 fair

**Summary:**

The paper discusses the relation between the optimization landscape around the true solution and the depth of deep neural networks. They theoretically prove that a simple sub-gradient method (SubGM) converges to a neighborhood of the balanced true solution whose radius is determined by the depth N of the model via an N-layer linear neural network with noise. They also experimentally verify their conclusion on ResNet of CIFAR-10.

**post-rebuttal**

The rebuttal answers most of my questions and I decide to change my score. But I think the author should claim that the focus on diagonal linear model in the main boy to avoid confusion.

**Questions:**

Same as the Weakness.

**Limitations:**

The authors adequately addressed the limitations and potential negative societal impact of their work

**Strengths And Weaknesses:**

Strengths:
1. Theoretically prove the relation between optimization landscape/generalization error and the depth of the neural network.

Weakness:
1. A little bit curious about the statement "deeper models generalize better" in line 18. In my understanding, the too deep model will cause the problem of overfitting, especially without the residual block. Could the author explain this statement?

2. The structure of the linear neural network seems to be incorrect. The author uses the Hadamard product when defining the N-layer linear neural network. Take 2-layer linear network and $x\in\mathbb{R}^2$ as an example. The output of the network is defined as $y=<w_1⊙w_2 ,x>= w_{11}w_{21}x_1+w_{12}w_{22}x_2$. So here comes the problem: what is the number of neurons in the hidden layer? The structure of the linear model that the proof of the whole paper based on is really strange and I am curious whether the conclusion drawn in the paper is correct.

3. The author only considers the linear network without the activation function. However, the nonlinear activation function is quite important in deep learning to enhance the expression ability of models in more general and sophisticated application scenarios. I am curious does the performance of deep learning models consists with the theory proven on the linear models? The author only provides the generalization error performance on 2-4 layers simple neural network and I don't think it is convincing enough. In appendix B.3, I only find the test and train accuracy of Resnets and do not find the generalization error performance of Resnets. I think the author should do more experimental research.

---

> ### Author Response · Authors · 2022-07-31
> **Response to Reviewer ffkC for Weaknesses 1, 2**
>
> We thank the reviewer for the comments, which are addressed below.
>
> ### Response to Weaknesses
>
> >  "A little bit curious about the statement "deeper models generalize better" in line 18. In my understanding, the too deep model will cause the problem of overfitting, especially without the residual block. Could the author explain this statement?"
>
> We apologize for the confusion. We do not claim that "deeper models generalize better"; instead, this statement is meant to highlight the recent empirical success of deep models in modern learning tasks. The better generalization of deeper models is observed and reported in several recent papers. For instance, the recent paper [1] starts off by mentioning "In practice it is often found that large over-parameterized neural networks generalize better than their smaller counterparts, an observation that appears to conflict with classical notions of function complexity, which typically favor smaller models." The authors verify this claim by conducting experiments on thousands of models with various fully-connected architectures, optimizers, and hyper-parameters. As another example, the flagship paper [2] makes a similar remark: "Despite their massive size, successful deep artificial neural networks can exhibit a remarkably small gap between training and test performance."; see also the recent vision paper [3].
>
> >  "The structure of the linear neural network seems to be incorrect. The author uses the Hadamard product when defining the N-layer linear neural network. Take 2-layer linear network and $x\in\mathbb{R}^2$ as an example. The output of the network is defined as $y = \langle w_1\odot w_2, x\rangle = w_{11}w_{12}x_1+w_{21}w_{22}x_2$. So here comes the problem: what is the number of neurons in the hidden layer? The structure of the linear model that the proof of the whole paper based on is really strange and I am curious whether the conclusion drawn in the paper is correct."
>
> As mentioned throughout the paper, we consider the deep linear regression problem, where the true parameter is modeled as a *diagonal linear neural network*. Our model is a special case of a general linear neural network of the form $y = W_0W_1\dots W_N x$, where $W_0\in \mathbb{R}^{1\times d}$ is the vector of ones and the square weight matrices $W_1,\cdots,W_N$ are restricted to be diagonal (hence resulting in the hadamard product). In light of this, the number of neurons in each hidden layer is equal to $d$ (in the example provided by the reviewer, the number of hidden neurons in the hidden layer is 2). We would also like to point out that the diagonal linear NNs has been extensively studied in the literature, see e.g. \[4-7\] for 2-layer diagonal linear NNs and \[8-10\] for N-layer diagonal NNs.
> The reviewer claims that "the structure of the linear neural network seems to be incorrect" and "conclusion drawn in the paper is incorrect". We kindly ask the reviewer to clarify which parts of our arguments are suspected of being incorrect. We are looking forward to addressing any remaining concerns regarding our results.
>
> [1] Novak, Roman, et al. "Sensitivity and generalization in neural networks: an empirical study." arXiv preprint arXiv:1802.08760 (2018).
>
> [2] Zhang, Chiyuan, et al. "Understanding deep learning (still) requires rethinking generalization." Communications of the ACM 64.3 (2021): 107-115.
>
> [3] Bommasani, Rishi, et al. "On the opportunities and risks of foundation models." arXiv preprint arXiv:2108.07258 (2021).
>
> [4] Tomas Vaskevicius, Varun Kanade, and Patrick Rebeschini. Implicit regularization for optimal sparse recovery. *Advances in Neural Information Processing Systems*, 32:2972–2983, 2019.
>
> [5] Blake Woodworth, Suriya Gunasekar, Jason D Lee, Edward Moroshko, Pedro Savarese, Itay Golan, Daniel Soudry, and Nathan Srebro. Kernel and rich regimes in overparametrized models. In *Conference on Learning Theory*, pages 3635–3673. PMLR, 2020.
>
> [6] Peng Zhao, Yun Yang, and Qiao-Chu He. Implicit regularization via hadamard product over- parametrization in high-dimensional linear regression. *arXiv preprint arXiv:1903.09367*, 2019.
>
> [7] Jeff Z HaoChen, Colin Wei, Jason Lee, and Tengyu Ma. Shape matters: Understanding the implicit bias of the noise covariance. In *Conference on Learning Theory*, pages 2315–2357. PMLR, 2021.
>
> [8] Li, Jiangyuan, et al. "Implicit sparse regularization: The impact of depth and early stopping." *Advances in Neural Information Processing Systems* 34 (2021): 28298-28309.
>
> [9] Daniel Gissin, Shai Shalev-Shwartz, and Amit Daniely. The implicit bias of depth: How incremental learning drives generalization. *arXiv preprint arXiv:1909.12051*, 2019.
>
> [10] Blake Woodworth, Suriya Gunasekar, Jason D Lee, Edward Moroshko, Pedro Savarese, Itay Golan, Daniel Soudry, and Nathan Srebro. Kernel and rich regimes in overparametrized models. In *Conference on Learning Theory*, pages 3635–3673. PMLR, 2020.

---

> > ### Comment · Reviewer_ffkC · 2022-08-09
> > **Thanks for the response.**
> >
> > The rebuttal answers most of my questions and I decide to change my score to 5. But I think the author should claim that the focus on the diagonal linear model in the main boy to avoid confusion.

---

> ### Author Response · Authors · 2022-07-31
> **Response to Reviewer ffkC for Weakness 3**
>
> ### Response to Weaknesses 3
>
> > "The author only considers the linear network without the activation function. However, the nonlinear activation function is quite important in deep learning to enhance the expression ability of models in more general and sophisticated application scenarios. I am curious does the performance of deep learning models consists with the theory proven on the linear models? The author only provides the generalization error performance on 2-4 layers simple neural network and I don't think it is convincing enough. In appendix B.3, I only find the test and train accuracy of Resnets and do not find the generalization error performance of Resnets. I think the author should do more experimental research."
>
> We thank the reviewer for this comment. Needless to say, in order to study the optimization landscape of the most sophisticated deep models, it is imperative to first have a complete understanding of simpler models. The effect of depth on the optimization landscape of deep models has remained elusive to this day, even for diagonal linear models. To bridge this knowledge gap, we consider the robust learning of deep linear regression, which has a crisp mathematical formulation and can serve as a test bed for more sophisticated models.
> The reviewer claims that our simulations are not convincing enough. With all due respect, we have to disagree with the reviewer: we believe that our simulations on deep diagonal linear models are completely in line with our theoretical analysis; we kindly refer to our motivating example in page 2, our discussions after Theorems 2 and 3 (where we clarified the connection between these theorems and Figure 1), and sections B1 and B2 in the appendix. Moreover, we never claimed that our theoretical analysis readily extends to more general nonlinear models. Instead, through extensive simulations on realistic ResNet models and CIFAR datasets, we have provided strong empirical evidence suggesting that similar phenomena are also prevalent in nonlinear models. We believe that these observations call for a deeper theoretical analysis of the optimization landscape of deep nonlinear neural networks; a goal that is the core of our future research.
> Finally, we are unsure what the reviewer means by "I only find the test and train accuracy of Resnets and do not find the generalization error performance of Resnets." The generalization error can be readily obtained as one minus the test accuracy, which is already reported in the appendix.

---

### Official Review · Reviewer_x9hu · 2022-07-11

**Rating:** 8
**Confidence:** 4
**Soundness:** 3 good
**Presentation:** 4 excellent
**Contribution:** 3 good

**Summary:**

This paper studies an interesting problem called "grossly corrupted linear model", where the ground truth is a linear model and a portion of output samples corrupted with noises with considerable variances. A diagonal linear network with depth N is used to learn the model, and the authors prove/propose several interesting results:
1) There exist parameters that correspond to the ground truth, but are neither local-min nor global-min.
2) The authors propose a sub-gradient optimization method, termed SubGM, that provably converges to the ground truth solution.
3) The deeper models have relatively better landscape around the ground truth solution. As the depth grows, SubGM indeed is more likely to converge to the ground truth solution.

Empirical results show that SubGM also works well on matrix recovery and Cifa-10 classification (with deep ReLU networks), and phenomenon 3) is also observed in these two problems.


**Questions:**

1. In the abstract, t would be better to mention that this paper only considers diagonal linear networks. I am not aware of this until I read Section 1.1.
2. The experiments on ReLU networks did not rule out the effect of representation power. In particular, the better performance of deeper models may be (partially) due to the stronger expressivity.
3. I feel "bless of depth" suits the title better. Once N>1, the model is no longer convex, but increasing depth still improves the performance of SubGM.

**Limitations:**

The author may want to discuss how to extend the results to realistic neural-net settings. So far the results are limited to diagonal linear models which are not commonly used in practice. I am curious about the main challenge of applying a similar analysis to fully-connected linear networks or a non-linear DNN.

**Strengths And Weaknesses:**

**Strength**:

The paper is clearly presented. It was believed that for neural networks, the depth increases the expressivity power, while the width helps optimization [1]. If we want to train very deep networks (say, more than 50 layers), certain tricks like skip connections and normalization must be applied. This paper, as far as I am concerned, provides a somewhat surprising insight: depth can also benefit the optimization landscape. With a proper learning algorithm, increasing the depth helps the optimization and the recovery of the ground truth.

This paper studies the landscape from some novel angles:
- (a) The solution we want (ground truth) is not necessarily a local-min or a global-min
- (b) Given (a), analyzing the local landscape around the solution we want, instead of the landscape around the global-min, can better help the optimization.

The paper establishes the theories based on the above ideas. It showed that the ground truth of the grossly corrupted linear model may not be the global-min of the optimization problem. Subsequently, it provides a landscape analysis to show that depth "smooths" the landscape around the ground truth solution. Then, it proposes an effective algorithm to recover the ground truth, with a convergence proof for any depth N≥2. Although the theory is not on neural-net settings, the experiments demonstrate that similar results can be observed in matrix recovery and image classification with neural networks.

Overall, this is a nice paper. It clearly sheds light on the optimization study of machine learning.

**Weakness**

The main weakness is that the diagonal linear models are not commonly used in practice. To me, it is a simplification of deep linear neural networks, which in turn is a simplification of fully-connected non-linear DNN. It is not clear whether the main theoretical conclusion (depth flattens the local landscape around the ground truth solution) still holds for practical settings.

[1] Sun et. al., The Global Landscape of Neural Networks: An Overview, arXiv:2007.01429.

---

> ### Author Response · Authors · 2022-07-31
> **Response to Reviewer x9hu**
>
> We are very happy that the reviewer finds our paper interesting.
>
> ### Response to Weaknesses
>
> > "The main weakness is that the diagonal linear models are not commonly used in practice. To me, it is a simplification of deep linear neural networks, which in turn is a simplification of fully-connected non-linear DNN. It is not clear whether the main theoretical conclusion (depth flattens the local landscape around the ground truth solution) still holds for practical settings."
>
> We thank the reviewer for this very insightful comment.
> Needless to say, in order to study the optimization landscape of the most sophisticated deep models, it is imperative to first have a complete understanding of simpler models. The effect of depth on the optimization landscape of deep models has remained elusive, even for diagonal linear models. To bridge this knowledge gap, we consider the robust learning of deep linear regression, which has a crisp mathematical formulation and can serve as a test bed for more sophisticated models. Admittedly, our results do not readily extend to more general settings, but through extensive simulations on realistic ResNet models and CIFAR datasets, we have provided strong empirical evidence suggesting that similar phenomena are also prevalent in nonlinear models; this indeed calls for a more rigorous study of deep networks from optimization and algorithmic perspectives.
> Moreover, as the reviewer correctly pointed out, in deep models, it is not clear how to disentangle the representation power of the NN from the properties of its optimization landscape. In diagonal linear NNs, the representation power of model remains unchanged with depth, since they all correspond to a model that is linear with the input. This in turn helps us study the optimization landscape in isolation by controlling the representation power.
>
> ### Response to questions
>
> >  "In the abstract, t would be better to mention that this paper only considers diagonal linear networks. I am not aware of this until I read Section 1.1."
>
> We thank the reviewer for the suggestion and apologize for the lack of clarity. We will mention in the abstract that we focus on the diagonal linear networks.
>
> >  "The experiments on ReLU networks did not rule out the effect of representation power. In particular, the better performance of deeper models may be (partially) due to the stronger expressivity."
>
> We agree with the reviewer that our comparisons for deep nonlinear models are slightly unfair since deeper models have larger expression power. However, we would like to argue that this effect is likely to be mild (at least in our simulations on ResNets with CIFAR dataset) since the used datasets are small compared to the capacity of the networks. For example, it is known that ResNet-18, ResNet-34, and ResNet-50 can memorize all the training data in CIFAR and achieve $100\%$ training accuracy. This in turn suggests that the expression power of these networks remains unchanged on CIFAR dataset. We will clarify these points in the revised paper.
>
> >  "I feel "bless of depth" suits the title better. Once $N>1$, the model is no longer convex, but increasing depth still improves the performance of SubGM."
>
> We agree with the reviewer that this is a much better title! We will modify the title in the revised paper.

---

> > ### Comment · Reviewer_x9hu · 2022-08-09
> > **Thanks for the response.**
> >
> > I totally understand that directly analyzing the landscape of a practical DNN is extremely challenging, and a diagonal linear model would be a good starting point as long as the insight can be verified through experiments. As for the issue of representation power, I think the authors also made a fair justification. I have no further concerns. Thanks again for the authors' effort.

---

### Official Review · Reviewer_9hKR · 2022-07-14

**Rating:** 6
**Confidence:** 3
**Soundness:** 3 good
**Presentation:** 3 good
**Contribution:** 3 good

**Summary:**

The paper studies the training dynamics of deep linear regression models. Specifically, it explores the effect of depth on the loss landscape of robust deep linear models. The results are mainly in two aspects: First, there is a phase transition in the loss landscape of the robust linear regression problem: when the dimension of the feature is much larger than the number of data samples, there exists a true solution that is not a critical point of the $\ell_1$-loss; otherwise, all true solutions are global minima. Second, sub-gradient methods converge to a global minimum that is close to the true solution, and moreover, depth slows down the training rate but improves the generalization error.

**Questions:**

1. It seems that the feasible region of $m$ in Theorem 1 and Theorem 2 intersects. Does that mean that subgradient methods will find a solution that has exactly $O(d)$ loss?

2. Is there any result showing that training rate and generalization error is directly correlated with the depth $N$?

3. In Line 174, it is mentioned that the improvement is significant if $m$ is small. How large is this $m$? It seems that by the assumption in Theorem 3, $m$ is not that small.

**Limitations:**

Yes.

**Strengths And Weaknesses:**

The paper is technically sound. The characterization of the phase transition in the landscape of deep linear models looks interesting, especially as RIP does not work in this case. The technique of the analysis may also be potentially applied to other settings.

I am a bit confused about the main information this paper wants to convey. In my humble opinion, the main information is that: for robust deep linear models, some true solutions may not be obtained by first-order methods, but sub-gradient method obtains a solution that is close enough. This information is not that related to the depth. The information related to depth is: depth flattens the local landscape around the true solution, slows down the training rate and improves the generalization error. The latter two points are obtained by comparing the results of two-layer and (more-than-)three-layer networks, which seems a little strange: it does not claim that adding depth after depth exceeds three still improves the generalization error, so I'm not sure whether depth plays a role here.

---

> ### Author Response · Authors · 2022-07-31
> **Response to Reviewer 9hKR for Weaknesses**
>
> We are glad that the reviewer finds the results of our paper interesting.
>
> ### Response to Weaknesses
>
> > "I am a bit confused about the main information this paper wants to convey. In my humble opinion, the main information is that: for robust deep linear models, some true solutions may not be obtained by first-order methods, but sub-gradient method obtains a solution that is close enough. This information is not that related to the depth. The information related to depth is: depth flattens the local landscape around the true solution, slows down the training rate and improves the generalization error. The latter two points are obtained by comparing the results of two-layer and (more-than-)three-layer networks, which seems a little strange: it does not claim that adding depth after depth exceeds three still improves the generalization error, so I'm not sure whether depth plays a role here."
>
> We appreciate your insightful comment and apologize for the confusion about the main contributions of our paper. We would like to point out that the main overarching message of our paper is that *depth can benefit the optimization landscape of deep linear regression*. We start off by showing that the true solution may not be a global minimum of the loss function, but for deeper models the true solution emerges as a local minimum. This motivated us to study the effect of depth on the optimization landscape around this local solution, as opposed to the global minimum (as pointed out by Reviewer x9hu). First, we show that depth can flatten the optimization landscape around the true balanced solution: by going from $N$ to $N+1$ layers, the landscape is flattened by at least *a constant factor*. Second, we show that depth affects both the convergence rate and escape time of the algorithm. In particular, deeper models take longer to train, but once trained, the algorithm will stay close to the ground truth for a longer time. Moreover, we precisely characterize the role of depth on the convergence rate of the subgradient method, showing that it consistently slows down with depth (from $O(1/\epsilon)$ for 2-layer models to $O(1/\epsilon^2)$ for infinitely deep models). Indeed, our results hold for *any* depth $N\geq 2$.
>
> Moreover, we would like to clarify the effect of depth on the generalization error. Our Theorems 2 and 3 show that the generalization error undergoes two major improvements as depth increases from 1 to 2, and from 2 to 3 or more. Our theoretical results do not show further improvement in the generalization error beyond 3-layer models. We believe that this is *not* an artifact of our technique, as our extensive numerical experiments confirm that SubGM yields similar generalization error on 3- to 6-layer models (see Figure 1 in the paper and Figure 3 in the appendix). We will certainly clarify this point in the revised manuscript.

---

> > ### Comment · Reviewer_9hKR · 2022-08-09
> > **Thanks for the detailed response. Raised my score to 6.**
> >
> > Thank you for your detailed response. I am now convinced that the paper does convey a message that depth plays a role in the optimization of neural networks. Thus, I have raised my score to 6.

---

> ### Author Response · Authors · 2022-07-31
> **Response to Reviewer 9hKR for Questions**
>
> ### Response to questions:
>
> > "It seems that the feasible region of $m$ in Theorem 1 and Theorem 2 intersects. Does that mean that subgradient methods will find a solution that has exactly $O(d)$ loss?"
>
> We thank the reviewer for this insightful comment. Indeed, the feasible region for $m$ in Theorems 1 and 2 *do* intersect. To explain this, we point out that for $N$-layer models with $N\geq 2$, there are infinitely many true solutions (the size of the set $\mathcal{W}$ is infinity). Theorem 1 shows that, independent of depth and as long as $m\leq 0.1d$, there *exists* at least one true solution from the set $\mathcal{W}$ that is not a stationary point. On the other hand, Theorem 2 shows that for models with depth $N\geq 2$ and $m\gtrsim k^2$, SubGM converges to a *specific* balanced solution (different from the hidden solution in Theorem 1). Combined together, these two statements imply that in the regime where $k^2\lesssim m\leq 0.1d$, deep linear regression has *both* "bad" (i.e. hidden and unidentifiable) and "good" (i.e., recoverable via SubGM) true solutions. We will further clarify this point in the paper.
>
> > "Is there any result showing that training rate and generalization error is directly correlated with the depth $N$?"
>
> Based on our understanding, the reviewer is using "training rate" to refer to the *training time* (please let us know if our interpretation is wrong). There is indeed a close relationship between the training time, generalization error, and depth. As mentioned above, deeper models take longer to train, but once trained, they have a more stable behavior around the ground truth. On the other hand, the generalization error undergoes two transition phases as depth increases from 1 to 3 or more. We kindly refer the reviewer to our response to Weaknesses.
>
> >  "In Line 174, it is mentioned that the improvement is significant if $m$ is small. How large is this $m$? It seems that by the assumption in Theorem 3, $m$ is not that small."
>
> We thank the reviewer for this clarifying question. Note that the required lower bound on $m$ in Theorem 3 is nearly dimension-free (it only has logarithmic dependency on $d$). More specifically, ignoring the dependency on the logarithmic factors, corruption probability, and the condition number, we only require $m\gtrsim k^2$, which can be significantly smaller than the dimension $d$. In light of this lower bound, the improvement from 2- to 3- (or more) layer models is in the order of $O((1/\alpha)^{\Omega(1)})$, which is significant for small choices of $\alpha$.

---

### Meta-Review · Area_Chair_xWFz · 2022-08-26

**Recommendation:** Accept
**Confidence:** Less certain

**Metareview:**

This paper studies a linear network for a regression problem. The main objective is to provide more understanding of the optimization landscape and characterizes the effect of the structure (depth) on the neural network in the oper-parametrized setting. The paper balance and shows both encouraging results but also addresses the weaknesses of the optimization landscape. The paper is concluded with interesting numerical experiments that support the claims in the paper. Let me also highlight that the theoretical analysis includes many novel parts (in the appendix).


**Award:**

No

---

### Decision · Program_Chairs · 2022-09-14

Accept